# Effects of Food Concentration and Light Intensity on the Growth of a Model Coral

**Tung-Yung Fan [1,2,*]**, **Yan-Leng Huang [1]** and **Anderson Mayfield [3,4,5,*]**

1    National Museum of Marine Biology & Aquarium, Pingtung 944, Taiwan
2    Department of Marine Biotechnology & Resources, National Sun Yat-sen University, Kaohsiung 804, Taiwan
3    Coral Reef Diagnostics, Miami, FL 33129, USA
4    International Coral Reef Society, Tavernier, FL 33070, USA
5    Coral Research and Development Accelerator Platform, Thuwal, Makkah 23955, Saudi Arabia
*    Correspondence: tyfan@nmmba.gov.tw (T.-Y.F.); anderson@coralreefdiagnostics.com (A.M.)

**Abstract:** Since reef-building corals rely on both heterotrophy and endosymbiotic dinoflagellate autotrophy to meet their metabolic needs, it is necessary to consider both food supply and light levels, respectively, when optimizing their cultivation *ex situ*. Herein nubbins of the model reef coral *Pocillopora acuta* cultured in recirculating aquaculture systems at photosynthetically active radiation levels of 370 or 670 µmol quanta m$^{-2}$ s$^{-1}$ were fed *Artemia* nauplii at concentrations of either 33 or 78 individuals mL$^{-1}$ in a separate feeding tank for 6 hr in the dark thrice weekly. A subset of nubbins was experimentally wounded at the outset of the 84-day experiment to assess recovery, and 100% fully healed within 2–4 weeks. All cultured corals survived, and unwounded corals (1) grew at a specific growth rate approaching 0.5% day$^{-1}$ and (2) demonstrated a mean total linear extension of 0.2% day$^{-1}$ (~6–8 cm year$^{-1}$); these are far higher than growth rates normally documented *in situ*. In the feeding tank, corals tolerated nitrate levels up to 25 mg L$^{-1}$, but once concentrations reached 50 mg L$^{-1}$ by day 84, tissue necrosis began to occur in nubbins of one tank. This highlights the importance of feeding in separate tanks during long-term culture of corals, and bio-filtration could reduce the possibility of organic matter accumulation in future coral culture studies.

**Keywords:** aquaculture; coral reefs; heterotrophy; husbandry; recirculating aquaculture systems





## 1. Introduction

As coral reefs continue to decline due to ocean warming, pollution, overfishing, and other factors [1], *ex situ* coral husbandry and propagation are becoming increasingly important conservation tools [2]. Although it is still more common to culture corals in aquarium facilities to later sacrifice them in manipulative experiments [3,4], there has also been an interest in cultivating corals *ex situ* to establish a sustainable supply of biological material for ecosystem restoration, the aquarium trade, research [5], and other industries (*e.g.*, natural products [6,7]). The recirculating aquaculture system (RAS) has proven to play an important role in coral culture [8,9], especially since they are relatively easy to maintain; seawater conditions can be readily adjusted and optimized within a RAS to promote coral growth and health [10,11] and even induce spawning [12]. Since corals house photosynthetic dinoflagellates of the family Symbiodiniaceae, from which they receive a significant amount of nourishment in the form of fixed organic carbon, proper optimization of the light environment is critical for successful coral culture [13–15].

Coral heterotrophy must also be considered, and the use of "live rock" [16,17] alongside active feeding regimens can improve coral health *ex situ* [18]. Coral heterotrophy is well studied [19–22] and can become even more important in corals that have lost their ability to photosynthesize as a result of environmental stress [23]. Despite the importance of heterotrophy, it is oftentimes omitted from aquarium studies for a number of reasons, one being that it can result in eutrophication within the tanks [24]. In a recent study with

*Pocillopora acuta*, however, we showed that, by feeding corals *Artemia* nauplii in a separate "feeding tank" during long-term husbandry studies, this issue is avoided [9]. In the same study, we found that feeding is critical to maintaining healthy pocilloporids in culture, as has been documented by others [25].

We hypothesized that we could build upon these advancements, which included amongst the highest coral growth rates ever measured (~6 cm total linear extension [TLE] per year) by using photosynthetically active radiation (PAR) levels more representative of shallow (3–5 m), clear-water habitats [26,27]: ~370 and 670 µmol (quanta) $m^{-2} s^{-1}$ vs. a maximum (max.) of 250 µmol $m^{-2} s^{-1}$ in [9]. We also found that a flow of 6 cm $s^{-1}$ led to faster coral growth than at 4 cm $s^{-1}$ [9]; herein we instead generated both forward and backward flow at ~5 cm $s^{-1}$ (lower than used with larger-polyped corals [28]) at each PAR level. Since we previously documented flow effects [9], we sought to test the effects of two PAR levels and two food regimes on *P. acuta* physiology. Our prior work featured *Artemia* concentrations in the feeding tank of ~50 *Artemia* $mL^{-1}$; herein we chose one value slightly lower than this (33 *Artemia* $mL^{-1}$) and one higher (78 *Artemia* $mL^{-1}$). We hypothesized that corals cultured at the higher of the two PAR levels would grow faster based on the shallow depths from which the corals were collected (~3–5 m). Likewise, we hypothesized that corals would perform better at the higher of the two food regimes. The overarching goal was to continue to optimize the culture conditions for this important model coral [29,30].

Coral propagation typically involves cutting large colonies into small nubbins, though the resulting lesions represent a potential entry point for pathogens [31,32], and various fouling organisms can thwart coral recovery [33], growth [34], and reproduction [35]; recovery can take weeks or months, is species-dependent, and also varies across intrinsic physiological properties of coral colonies (as well as environmental factors [36,37]). Heterotrophic feeding can facilitate wound recovery [38], and, as a secondary goal of this work, we sought to identify the optimal combination of light and food that would result in superior wound healing in our Taiwanese *P. acuta* specimens.

## 2. Materials and Methods

### 2.1. Biological Material

Six *P. acuta* colonies (diameter = ~12–15 cm) were collected as described previously [9] at 3–5 m, quarantined in the husbandry facility of the National Museum of Marine Biology and Aquarium for two weeks, and acclimated in a semi-transparent, semi-enclosed, 30-ton flow-through tank with sand-filtered (~50 µm) natural seawater at 26 ± 1 °C (mean ± standard error for this & all other error terms unless stated otherwise), a salinity of 35 ± 1, and 160–450 µmol $m^{-2} s^{-1}$ over a 12/12 hr light/dark cycle. Afterward, 13–14 nubbins were cut to ~2 ± 1 cm (max. length) with a scalpel from each colony, and the 80 nubbins were glued (Ista, Tainan, Taiwan) to 2.7-cm, T-shaped ceramic pedestals (Oceanexus, Tainan, Taiwan) and allowed to recover for four weeks under the conditions described above. Two experiments, both outlined below, were undertaken using the same light and food regimes in a split-plot design; 60 nubbins were cultured for 120 days without further wounding beyond creating the initial nubbin ("growth experiment"), with the remaining 20 used in a "wound-healing experiment" that took place simultaneously (albeit with data collected only over 42 days).

### 2.2. Wound-Healing Experiment

Twenty of the eighty nubbins were wounded by cutting off a portion of the nubbin using bone-cutting pliers three days after the initial four-week acclimation period as per a standardized protocol [38] and monitored over six additional weeks at each of the four culture conditions outlined below; note that the full gamut of physiological response variables (also described below) was not assessed in the wounded nubbins. Samples were photographed using a Leica M165FC stereo-microscope (Wetzlar, Germany) immediately after wounding, as well as after two and four weeks of incubation at the respective treatment condition ($n$ = 4–6 corals treatment$^{-1}$). The magnification and angle of the photograph were kept constant for each colony across all photographs, which were used to assess

wound recovery. Each colony was assigned to one of four recovery stages based on a qualitative assessment of the wound site [39]: (a) "incomplete occlusion" (wound remaining open or having increased in surface area, with bare skeleton still exposed; score = 0); (b) "full occlusion" (undifferentiated tissue covering a portion or entirety of the wound; score = 1); (c) "tentacle eruption" (presence of tentacles incapable of contraction and prey capture; score = 1.5); or (d) "full recovery" (emergence of a fully functional polyp capable of feeding at the wound site; score = 2).

## 2.3. The Culture System

The RAS was described in detail in [9]. Briefly, each of the three RASs included an upper culture tank (125 × 60 × 70 cm) connected to a lower "life support" tank (80 × 45 × 45 cm). The culture tank included "live rock" (25 kg) and LED lights (HLG-480H-C2100B, TME, Lodz, Poland) programmed to administer $672 \pm 33$ μmol m$^{-2}$ s$^{-1}$ ("high light" [HL]) in one region of each tank and $369 \pm 22$ μmol m$^{-2}$ s$^{-1}$ ("low light" [LL]) in the second half over a 12/12 hr light/dark cycle. A plastic divider was placed between the lamps, though since it did not enter the water (to avoid interfering with the flow), it is possible that light from one light region of the tank entered the other; for this reason, it was important to randomly rotate nubbins regularly within each tank light region to ensure that those on the low–high-light border did not experience significantly different PAR regimes from other nubbins within the same treatment. Each experimental tank also contained two GP-03 flow motors (Maxspect, Shenzhen, China), and corals were exposed to bidirectional flow: $5.9 \pm 0.3$ cm s$^{-1}$ in the forward direction and $5.1 \pm 0.2$ cm s$^{-1}$ in the reverse (mean flow = 5.5 cm s$^{-1}$). Light intensity and flow velocity were measured by Li-Cor (LI-193SA, Lincoln, NE, USA) and Kenek (GR20/GR3T-2–20N, Musashino, Japan) meters, respectively. The life support tank contained a 0.2 mm filter bag, a protein skimmer (JNS, $CO_2$, Taiwan), "live sand" (3 kg), an automatic Mato-2009 RO bucket (Autoaqua, Hsinchu, Taiwan), a zeolite drum (JNS, DC-2), a 6,000 L/H primary pump (Mr. Aqua, Taichung, Taiwan), a CS072A-1 titration system (Johnlen, Taiwan; for measuring kH & concentrations of calcium [$Ca^{2+}$] & magnesium [$Mg^{2+}$]), a 350-W heater (Ista), and a C-1000 p chiller (Resun, Shenzhen, China). The temperature was maintained at $26 \pm 1$ °C by the self-regulating heaters and chillers, and salinity was maintained at 35 using a Mato-200 osmoregulator (Autoaqua).

Commercially available nitrifying bacteria (A-5 Pandora, NBL, Taiwan) were added to the live sand monthly. To ensure consistent water quality, the three experimental tanks were connected and operated in series for eight weeks before the experiment; they were separated into distinct culture systems once the experiment began. The synthetic seawater in the RAS (Red Sea Salts [Israel] mixed with reverse osmosis water) was changed biweekly (30 & 100 L for the culture & feeding tanks, respectively), and concentrations of nutrients (nitrate, nitrite, phosphate, & ammonium), $Ca^{2+}$, and $Mg^{2+}$, as well as kH and pH, were measured biweekly (Profi Test, Salifert, Holland). A more detailed treatise on seawater quality can be found in Supplementary File S1.

## 2.4. Light and Feeding Treatments

Each of the 60 nubbins of the growth experiment was randomly assigned a culture tank (1, 2, or 3; $n = 20$ nubbins tank$^{-1}$) and then randomly placed at one of the two aforementioned light levels (HL or LL); a 12/12 hr light/dark cycle was used for each of the three culture tanks, which featured 10 nubbins each at the two light levels. Half (randomly selected) of the 10 nubbins in each light region in each tank were fed $78 \pm 7$ two-day-old *Artemia nauplii* mL$^{-1}$ (hereafter *Artemia* mL$^{-1}$) for 6 hr ("high food" [HF]), whereas the other half were fed only $33 \pm 4$ *Artemia* mL$^{-1}$ ("low food" [LF]) for 6 hr. As such, each tank featured five nubbins at each of the four treatments: low-light + low-food (LLLF), low-light + high food (LLHF), high-light + low-food (HLLF), and high-light + high-food (HLHF; $n = 5$ nubbins treatment$^{-1}$ tank$^{-1}$ × 4 treatments × 3 culture tanks).

Two days before commencing feeding, 10 g of *A. salina* cysts (Supreme Plus, Golden West Artemia, Ogden, USA) were incubated in a 2-L, well-aerated hatching cone featuring artificial seawater for 48 hr at $27 \pm 1$ °C and a salinity of $35 \pm 1$. The nauplii were enriched by adding

1.5 mL of 100 ppm of a soluble commercial product (Pack Boost Enrichment Diets, Omega®, Chuan Kuan, Kaohsiung, Taiwan) 36- and 42-hr post-hatching. A magnetized cyst collector tube was used to remove the unhatched cysts or shells [40]. The two-day-old nauplii were collected by a 200-μm filter, rinsed with artificial seawater, and added to an independent feeding tank system that included an upper feeding tank ($120 \times 60 \times 60$ cm) connected to a lower life support tank ($80 \times 45 \times 45$ cm). The corals were fed while within a plankton net ($20 \times 55 \times 70$ cm), with bubble stones in the four corners to allow for even water mixing. After the lights were turned off for 30 min, all coral nubbins of the same culture tank × feeding regime were moved into the feeding tank for 6 hr, with small (~1–2 mL) water samples taken after 0, 3, and 6 hr to estimate how many nauplii were available for feeding (as well as to estimate the relative feeding rate). Coral nubbins were rinsed with seawater prior to return to their experimental tanks.

## 2.5. Buoyant Weight (BW), Specific Growth Rate (SGR), and TLE

The weights of the coral nubbins were measured by a BW technique [41] on a Mettler Toledo AB204 balance (precision = 0.0001 g; Columbus, USA) every two weeks (i.e., days 0, 14, 28, 42, 56, 70, 84, 98, & 112). A beaker containing filtered seawater ($26 \pm 0.5$ °C & salinity of 35) and a thermostatic bath were placed under the scale, and the nubbins were suspended on fishing line under the scale for BW measurements. Before each measurement, the surface of the coral pedestal was brushed with a toothbrush to remove algae. SGR (% day$^{-1}$) was calculated as follows: $((\ln(W_f) - \ln(W_i))/\Delta t \times 100)$, where $\ln(W_i)$ and $\ln(W_f)$ represent the natural logarithms of the nubbin BW (g) at the beginning and end of the experiment, respectively, and $\Delta t$ represents the duration in days. As another proxy for growth, vernier calipers were used to measure the max. length, width, and height (cm) of the nubbins. The resulting values were summed to yield TLE [42], and % increases over time were assessed.

## 2.6. Maximum Quantum Yield of Photosystem II ($F_v/F_m$)

$F_v/F_m$ was measured using pulse amplitude modulation (PAM) fluorometry (Diving PAM, Walz, Effeltrich, Germany). After turning off the lights for 30 min, both minimum ($F_o$) and maximum fluorescence ($F_m$) were measured for each nubbin, and $F_v/F_m$ was calculated as $F_v/F_m = (F_m - F_o)/F_m$. Measurements were made on each of the 60 and 20 nubbins of the growth and wound-healing experiments, respectively, every two weeks (as per BW measurements). Nubbin color was assessed as in Siebeck et al. [43]. However, since all non-wounded corals presented color values of 5 (the maximum) at all times, these data were not analyzed.

## 2.7. Statistical Analyses

Data were tested for normality (Shapiro–Wilk tests) and equal variance (Levene's tests), and, when necessary, Box–Cox transformations were undertaken to meet the criteria for parametric ANOVA. First, two-way, repeated measures (RM) ANOVAs were used to test for the effects of PAR level (df = 1), feeding regime (df = 1), and their interaction over time on the four response variables: BW, SGR, TLE, and $F_v/F_m$. Assessment time was the RM, and the nubbins were the repeated subjects. To accommodate the split-plot design, whereby multiple PAR levels were employed in the same tank, tank was nested within PAR as the split plot. Tank (treatment) was also assessed as a random effect. When an analysis was made at a single sampling time, a two-way split-plot ANOVA was instead used: light × food were the fixed factors, and the same random factors as above were included to accommodate the split plot. In certain cases, tank was also tested separately. Tukey's honest significant difference (HSD) tests were used to validate inter-mean differences, and an alpha of 0.01 was set for all statistical tests. All statistical analyses were undertaken with JMP® Pro 16 (JMP statistical discovery, Cary, NC, USA), and all data collected can be found in a Supplemental Excel File (i.e., Supplementary File S2).

## 3. Results

### 3.1. Seawater Quality

The light levels at the two tank regions were significantly different from one another and averaged 672 (HL) and 369 μmol m$^{-2}$ s$^{-1}$ (LL) over the duration of the study (Table 1). For more details on PAR, please see Figure 1A–C and Table S1 (Supplementary File S1). The flow regime was similar across tanks (Table 1), though varied over time (Table S1). The *Artemia* concentrations in the water were significantly higher for the HF treatment vs. the LF one (Table S1 and Figure 1D) and averaged 56 ± 2 (starting = 78 ± 7) and 13 ± 1 *Artemia* mL$^{-1}$ (starting = 33 ± 74; see Figure S2 [Supplementary File S1] for the data plotted over time.); food supply varied more significantly over time for the HF treatment (evident in Figure 1D). Other seawater parameters were generally similar among culture tanks (Table 1 and Table S1; Figure S1). The levels of ammonia, nitrite, nitrate, and phosphate in the three culture tanks were below detectable levels (<0.2 mg L$^{-1}$) during the entire experimental period. However, the nitrate concentration in the feeding tank rose over the course of the study, reaching 50 mg L$^{-1}$ by day 84 (Figure 2M). Several days afterward, tissue began sloughing off the skeletons for certain nubbins (Figure 2N–P). For this reason, we generally omitted data collected after day 90, though we tracked the degree of tissue necrosis for several weeks afterward.

**Table 1.** Comparison of seawater chemistry across tanks. Significant differences among tanks (Tukey's HSD, $p < 0.05$) are marked with lowercase letters, and error terms represent standard error of the mean. HL, LL, HF, and LF correspond to the high-light, low-light, high-food, and low-food treatments, respectively. To see statistical comparisons and data plotted over time, please see Table S1 and Figure S1, respectively (both in Supplementary File S1). Nitrite, ammonium, and phosphate concentrations were not measured at detectable levels in any culture tank. The flow in the feeding tank was uni-directional (*). Temp. = temperature.

| Parameter | HL | LL | HF | LF | Temp. | Salinity | Flow | NO$_3^-$ | pH | kH | Ca$^{2+}$ | Mg$^{2+}$ |
|---|---|---|---|---|---|---|---|---|---|---|---|---|
| Tank | μmol m$^{-2}$ s$^{-1}$ | | *Artemia* mL$^{-1}$ | | °C | unitless | cm s$^{-1}$ | mg L$^{-1}$ | unitless | | mg L$^{-1}$ | |
| 1 | 665 | 358 | 0 [b] | 0 [b] | 28.3 | 35.0 | 5.4 | 0 [b] | 8.21 | 6.46 | 477 [a] | 1351 [a] |
| 2 | 679 | 390 | 0 [b] | 0 [b] | 28.7 | 34.9 | 6.0 | 0 [b] | 8.20 | 6.56 | 474 [a] | 1258 [ab] |
| 3 | 673 | 360 | 0 [b] | 0 [b] | 28.7 | 34.8 | 5.1 | 0 [b] | 8.23 | 6.87 | 438 [b] | 1204 [b] |
| Feeding | 0 | 0 | 78 [a] | 33 [a] | 28.1 | 35.0 | 2.0 * | 22.2 [a] | 8.18 | 6.37 | 415 [b] | 1184 [b] |

### 3.2. Coral Response Variables

BW rose significantly over time in all treatments (Table 2 and Figure 3A–D), and no corals paled over the first 84 days of the experiment (color scores = 5 for all). The SGR also increased significantly over time (Table 2), though *post-hoc* differences over time were only documented in the HLLF group (Figure 3B), in which the SGR was significantly lower on day 14 vs. the final two assessment days. Although there were no treatment effects on BW or SGR (Table 2), SGR was highest overall in the HLLF (0.44% day$^{-1}$; Figure 3B); on average, samples grew from ~10 to ~15 g over 90 days (50% increase) in that treatment, with % BW increases of 33, 33, and 35% for the HLHF, LLHF, and LLLF treatments, respectively. In terms of TLE, corals of the HLHF, HLLF, LLHF, and LLLF treatments grew from 8.4, 8.1, 8.3, and 8.8 cm, respectively, on day 0 to 9.6, 9.4, 10.4, and 10.6 cm, respectively, on day 90 (Figure 4A–D; %TLE increases of 14, 16, 25, & 20%, respectively), and there was a statistically significant effect of time alone on TLE (Table 2); neither feeding regime nor light affected TLE (Table 2) or its percent increase. $F_v/F_m$ (Figure 5A,B) also changed significantly over time and did not vary across treatments (Table 2); global means of 0.72, 0.73, 0.73, and 0.74 were obtained for the HLHF, HLLF, LLHF, and LLLF treatments, respectively.

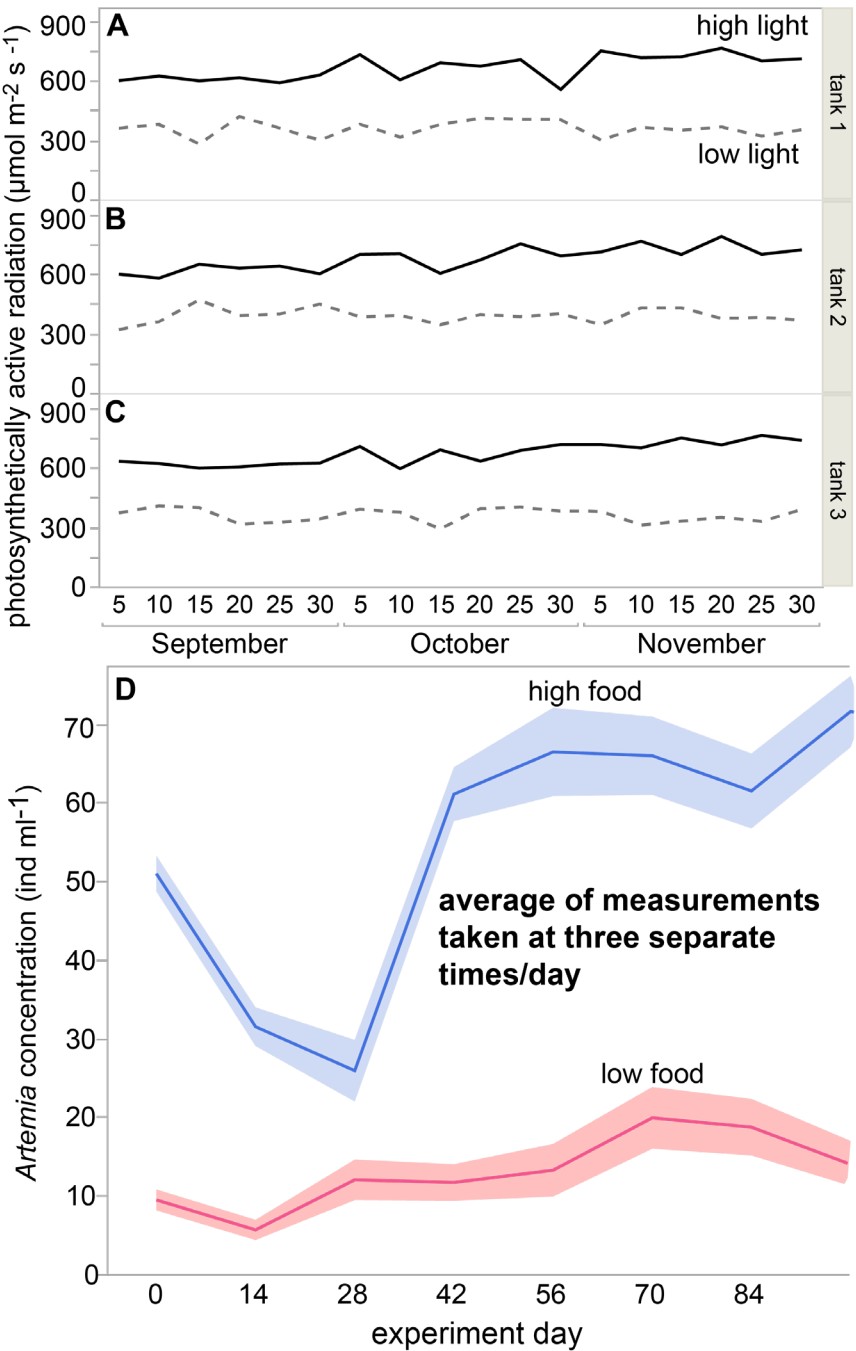

**Figure 1.** Experimental conditions. In the split-plot design, both high-light (solid black) and low-light (hatched gray) conditions were generated in each of three tanks (**A**–**C**), with mean photosynthetically active radiation levels shown over time for each treatment (tank). Corals (see Figure 2A–L for representative nubbin images.) were given either high (blue) or low (red) dosages of *Artemia* in a separate feeding tank (**D**) thrice per week, and solid lines and surrounding bands in (**D**) represent means and 95% confidence intervals, respectively. Note that these values are lower than the target initial concentrations of 33 and 78 *Artemia* mL$^{-1}$ (administered at time = 0 hr) since they reflect the means of the mean concentrations in the water over the duration of the 6-hr feeding period (see Figure S2 for details.).

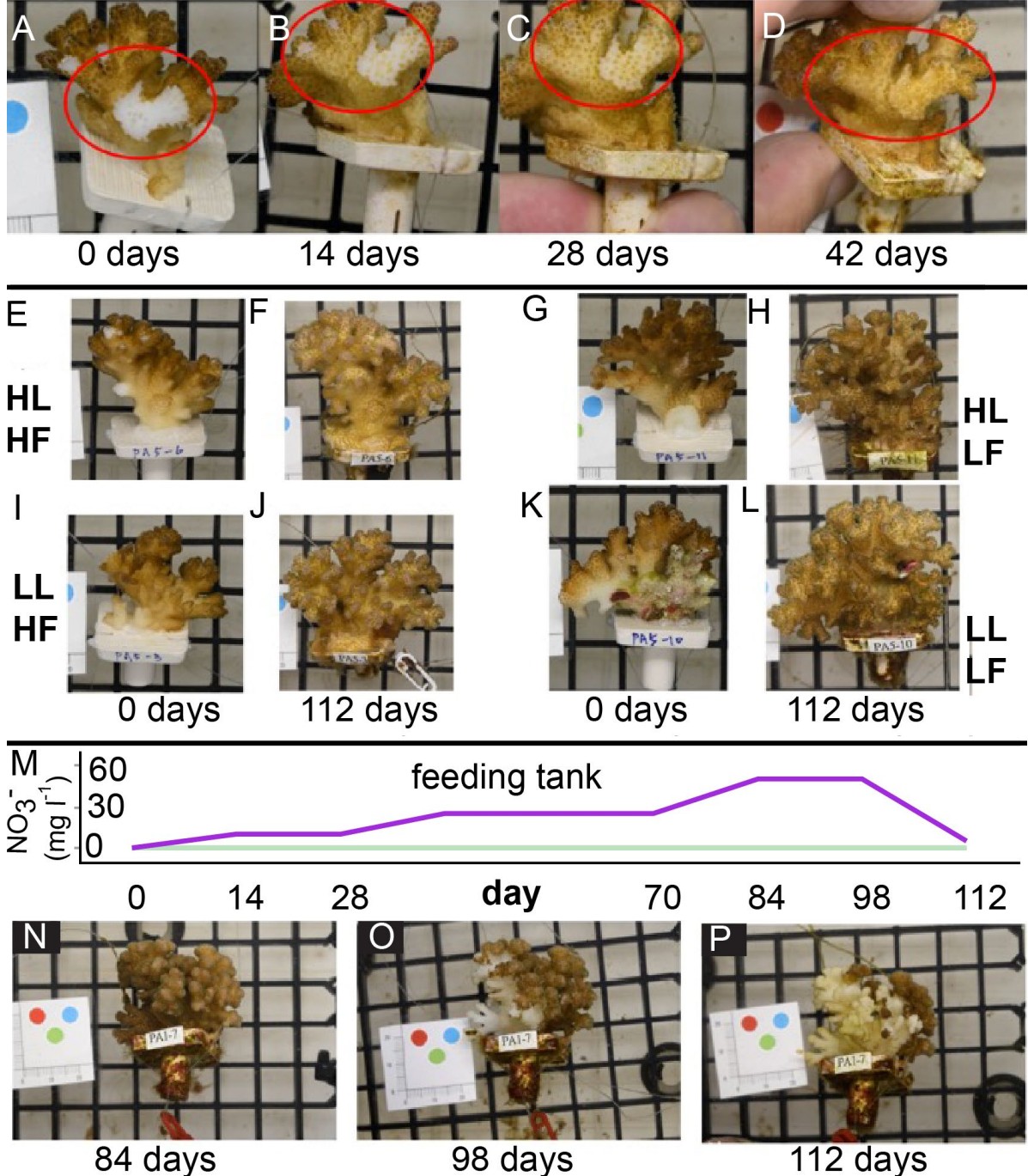

**Figure 2.** Representative images of coral nubbins. The representative images in panels (**A**–**D**) reflect scores of 0, 0.5, 1, and 2 from the wound-healing experiment, respepctively. Panels (**E**–**L**) depict representative images of corals of the high-light + high food (HLHF), high-light + low food (HLLF), low-light + high-food (LLHF), and low-light + low-food treatments (LLLF), respectively, at the beginning and end of the experiment. Panel (**M**) shows the nitrate concentrations in the feeding (purple) and culture (green) tanks over the duration of the experiment. Panels (**N**–**P**) show the progression of a representative nubbin that underwent tissue necrosis after immersion in the high-nitrogen feeding tank on day 84.

**Table 2.** Effects of light intensity and feeding regime on physiological performance of nubbins of the growth experiment (*n* = 60). Two-way repeated measures (RM) ANOVAs were carried out with nubbin as the repeated subject over time. Note that for measurements assessed on a single day (e.g., specific growth rate [SGR] on day 84), a standard two-way ANOVA (light × food) was instead undertaken.

| Source of Variation | df | *F* | *p* | *Post-Hoc* Comparisons |
|---|---|---|---|---|
| **Buoyant weight [a] (BW; g)—first 84 days** | | | | |
| light | 1 | 1.31 | 0.26 | |
| food | 1 | 0.08 | 0.77 | |
| light × food | 1 | 0.44 | 0.51 | |
| day | 8 | 129.74 | <0.01 | later > earlier |
| light × day | 8 | 0.63 | 0.70 | |
| food × day | 8 | 1.25 | 0.29 | |
| light × food × day | 8 | 0.32 | 0.94 | |
| **SGR [a] (% day$^{-1}$)—day 84 only** | | | | |
| light | 1 | 3.04 | 0.09 | |
| food | 1 | 1.70 | 0.20 | |
| light × food | 1 | 1.14 | 0.30 | |
| **SGR [a] —days 14–84 (RM ANOVA) [a]** | | | | |
| light | 1 | 1.87 | 0.18 | |
| food | 1 | 0.11 | 0.74 | |
| light × food | 1 | 1.24 | 0.27 | |
| day | 5 | 73.86 | <0.01 | later > earlier |
| light × day | 8 | 1.71 | 0.15 | |
| food × day | 8 | 2.04 | 0.09 | |
| light × food × day | 8 | 0.55 | 0.74 | |
| **Total linear extension (TLE; cm)—all times** | | | | |
| light | 1 | 2.21 | 0.14 | |
| food | 1 | 0.02 | 0.89 | |
| light × food | 1 | 0.46 | 0.50 | |
| day | 5 | 75.79 | <0.01 | later > earlier |
| light × day | 8 | 1.82 | 0.14 | |
| food × day | 8 | 0.35 | 0.84 | |
| light × food × day | 8 | 0.55 | 0.70 | |
| **TLE—first 60 days** | | | | |
| light | 1 | 0.90 | 0.35 | |
| food | 1 | 0.04 | 0.85 | |
| light × food | 1 | 0.35 | 0.56 | |
| day | 5 | 35.33 | <0.01 | later > earlier |
| light × day | 8 | 0.32 | 0.73 | |
| food × day | 8 | 0.62 | 0.54 | |
| light × food × day | 8 | 0.84 | 0.44 | |
| **TLE—% increase over 90 days** | | | | |
| light | 1 | 0.43 | 0.53 | |
| food | 1 | 0.03 | 0.88 | |
| light × food | 1 | 0.66 | 0.44 | |
| **Tissue sloughing (% area with necrotic tissue)** | | | | |
| light | 1 | 0.04 | 0.85 | |
| food | 1 | 0.04 | 0.85 | |
| light × food | 1 | 0.13 | 0.73 | |
| day | 5 | 7.91 | <0.01 | earlier > later |
| light × day | 8 | 0.73 | 0.39 | |
| food × day | 8 | 1.21 | 0.27 | |
| light × food × day | 8 | 0.13 | 0.71 | |
| **$F_v/F_m$ [a]** | | | | |
| light | 1 | 1.15 | 0.29 | |
| food | 1 | 1.29 | 0.26 | |
| light × food | 1 | 0.36 | 0.55 | |
| day | 8 | 17.93 | <0.01 | day-28 > day-42 |
| light × day | 8 | 1.12 | 0.35 | |
| food × day | 8 | 0.77 | 0.63 | |
| light × food × day | 8 | 1.53 | 0.15 | |

[a] Box–Cox-transformed data.

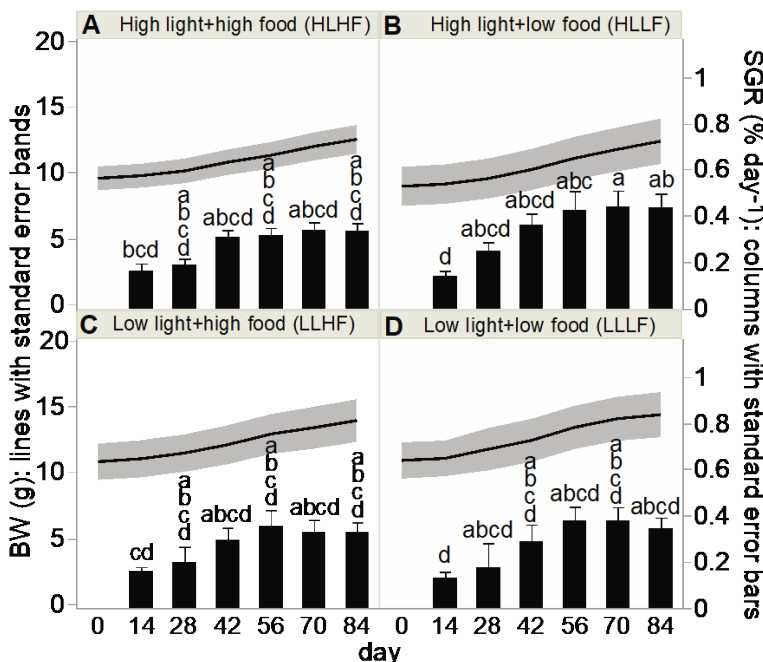

**Figure 3.** Buoyant weight (BW) and specific growth rate (SGR) of samples of the growth experiment ($n$ = 60). BW and SGR results are shown as solid black lines with gray standard error bands and black columns with standard error bars, respectively, for the HLHF, HLLF, LLHF, and LLLF treatments (panels (**A**–**D**), respectively). Temporal differences in the SGR are denoted by lowercase letters (Tukey's HSD $p < 0.05$), with statistical results of whole-model effects shown in Table 2.

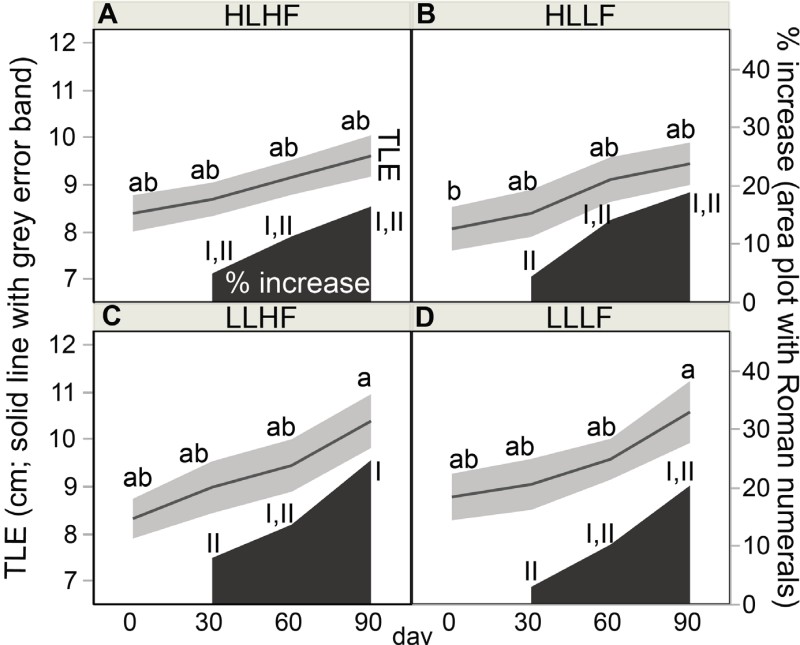

**Figure 4.** Total linear extension (TLE) of samples of the growth experiment ($n$ = 60). Raw TLE data (means) and percent increases are shown as black lines with gray standard error bands (left $y$-axes) and area plots (right $y$-axes), respectively. In all panels, lowercase letters denote significant temporal differences for left $y$-axis data (Tukey's HSD $p < 0.05$); Roman numbers instead denote significant differences in percent (%) TLE increase (right $y$-axes).

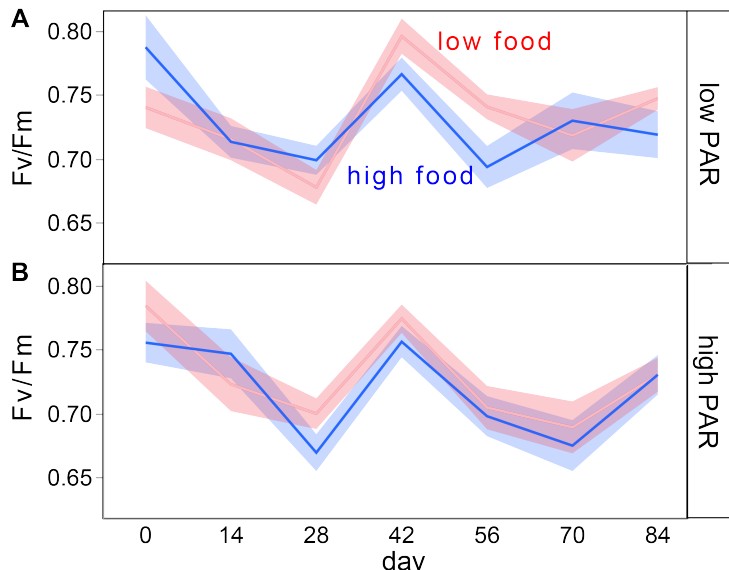

**Figure 5.** PAM fluorometry (Fv/Fm) data from samples of the growth experiment. $F_v/F_m$ data are presented as mean lines with standard error bands in panels (**A**,**B**) for the low- and high-light treatments, respectively; in each, the high- and low-food groups are represented by blue and red lines (with error bands), respectively.

Of the 20 nubbins that were experimentally wounded, 100% recovered in full by the 28th day (Figure 6A), and, in contrast to our hypothesis, there was no effect of treatment on wound healing or recovery speed (Table 2). Finally, tissue began sloughing off of the skeletons for certain nubbins around day 90 (Figure 2N–P and Figure 6B), just after nitrate levels reached 50 mg L$^{-1}$ in the feeding tank (Figure 2M). Significantly more nubbins succumbed to tissue sloughing in the second culture tank (*X*-squared, *p* < 0.001), and by day 112, over 75% of the tissues had detached from the skeleton in over half of the nubbins in tank 2.

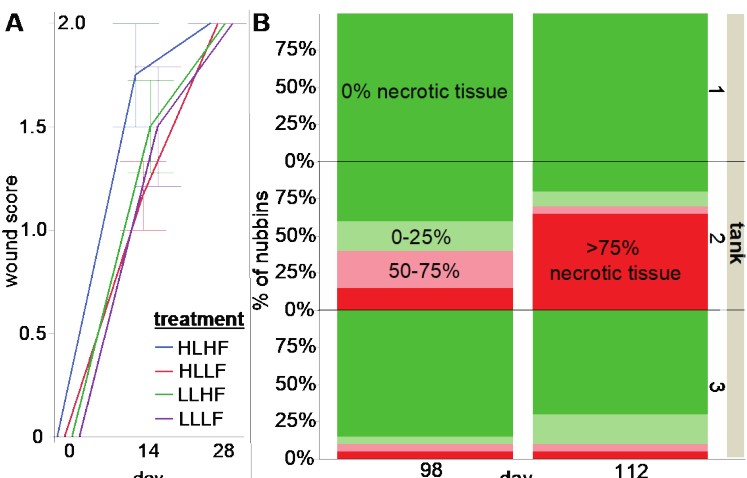

**Figure 6.** Wound healing and tissue necrosis. Wound scores from deliberately wounded nubbins (*n* = 20) are shown as raw mean wound score values ((**A**); error bars represent standard error of the mean). Day-42 data are not shown as all 20 nubbins remained fully healed from day 28 onwards. Panel (**B**) shows the distribution of samples from the growth experiment falling within each of four pre-defined bins on days 98 and 112: 0% necrotic tissue (dark green), 0–25% of surface area characterized by necrotic tissue (light green), 50–75% necrotic tissue (pink), and >75% necrotic tissue (dark red). No nubbins were characterized by 25–50% necrotic tissue, and a tank effect can be observed (see main text for $X^2$ results.).

## 4. Discussion

### 4.1. Coral Growth under Differing Light and Food Regimes

The average nubbin SGR of 0.4% day$^{-1}$ over 112 days represents a ~1.45-fold increase relative to starting mass and a doubling (in size) time of ~160–170 days. Upon excluding those corals that began exhibiting necrotic tissue around day 90, the mean SGR was nearly 0.5% day$^{-1}$. The highest growth rate of our cultivated nubbins occurred in the HLLF treatment (106 mg g$^{-1}$ week$^{-1}$), and, while ~50% higher than the means of the other treatments (76, 77, & 80 mg g$^{-1}$ week$^{-1}$ for HLHF, LLHF, & LLLF, respectively), this can only confidently be called a trend since, when conducting a simpler one-way ANOVA to test for the effect of treatment only, a *p*-value of 0.07 was obtained. These values are somewhat lower than global means from our prior work with this coral [9], in which the max. PAR was 250 µmol m$^{-2}$ s$^{-1}$ (with levels of 100 & 150 µmol m$^{-2}$ s$^{-1}$ also tested), and the overall mean SGR was nearly 0.6% day$^{-1}$. The food regimen was similar between studies, but a key difference is nubbin starting size: mean = ~10 g herein vs. 2–3 g in our prior work. As discussed below in the context of wound healing, smaller corals of the same species tend to exhibit higher growth rates, so this could explain the slightly higher growth rates in our prior study. Since culture at the higher of the two light levels (672 µmol m$^{-2}$ s$^{-1}$) employed herein did not lead to a significant increase in growth vs. 369 µmol m$^{-2}$ s$^{-1}$, ~250–370 µmol m$^{-2}$ s$^{-1}$ likely represents a recommended range for pocilloporids sampled from clear waters at 3–5 m. The use of higher PAR leads to no evident growth benefit and could even increase the chances of photo-bleaching, while lower levels (<~150 µmol m$^{-2}$ s$^{-1}$) impede calcification in other pocilloporids, even when fed [25]. A more detailed treatise on the factors that influence the growth of this coral species *ex situ* is provided below as part of a multi-study meta-analysis.

TLE increased on average by 23, 24, 30, and 27% over 120 days for the HLHF, HLLF, LLHF, and LLLF treatments, respectively (note differences vs. those presented over 90 days in the Results.). When looking at one dimension only (length), nubbins would, on average, be projected to grow 2.2, 2.2, 3.4, and 2.8 cm per year, respectively (TLE of 6.6, 6.5, 8.8, & 8.3 cm year$^{-1}$, respectively). These growth rates are similar to those of Osinga et al. [13], who also fed their pocilloporid corals *Artemia* nauplii, albeit at concentrations far higher than those used herein (2000 *Artemia* mL$^{-1}$) and alongside 30,000 *Tetraselmis suicica* cells mL$^{-1}$. Conlan et al. [44] documented lower growth rates than those documented herein (33% increase in BW vs. 50% herein), possibly because only 50 mg of *Artemia* nauplii (dry weight) were added each day to their 49-L culture tank. When relying on autotrophy alone (no supplemental feeding), Cunning et al. [45] observed *P. damicornis* growth rates approximately half of those measured herein; this highlights the importance of heterotrophy.

Despite wide variation in growth rates documented across studies, neither light nor the feeding regime significantly affected *P. acuta* growth rate herein. One reason for this could be that both PAR and food supply were already nutritionally saturating at the lowest levels employed: 360 µmol quanta m$^{-2}$ s$^{-1}$ and 33 *Artemia* mL$^{-1}$, respectively. Our similar growth rates to those of Osinga et al. [13], who used over 20 times higher food concentrations, could support this. That said, we tested aliquots of the water in the feeding tank throughout the 6-hr incubation period to estimate the feeding rate (Figure S2), and, in most instances of the low-food group, the water column was virtually depleted of *Artemia* by the end of feeding (88% decrease vs. only 45% decrease in the high-food group). In contrast, Tagliafico et al. [39] found that *Acropora millepora* and *Duncanopsammia axifuga* did not reach satiation until starting densities of 120 *Artemia* mL$^{-1}$ were used. Perhaps, then, the excess food consumption in the high-food treatments is converted into storage lipids, rather than growth-related processes. A future metabolomic exploration of these samples could be fruitful in potentially allowing us to understand how coral metabolism changes under saturated or even super-saturated food levels. Maybe well-nourished corals rich in lipid reserves would be better able to withstand future environmental change [2], as has been documented in a number of prior works (e.g., [46]). As a simpler explanation for

the lack of treatment effects observed, Schutter et al. [47] did not observe appreciable light effects on coral (*Galaxea* sp.) BW until around 180 days of culture with 10-fold different PAR levels (min = 38 & max = 410 μmol m$^{-2}$ s$^{-1}$, with growth significantly higher at the latter); it is possible, then, that light-related differences in *P. acuta* growth may ultimately have been documented were the experiment run for a longer period.

*4.2. Optimizing the Growth of P. acuta Ex Situ*

To take a further look into the conditions that led to maximized *ex situ* growth (as SGR) in our *P. acuta* genotypes, a meta-analysis was undertaken in which data from a prior work [9] were analyzed and compared to those obtained herein (Figure 7). In total, this resulted in four food regimes (0, 33, 43, or 78 *Artemia* mL$^{-1}$), five PAR levels (105, 157, 250, 370, or 670 μmol m$^{-2}$ s$^{-1}$), and two flow regimes: the 5 cm s$^{-1}$ bidirectional flow employed herein vs. the 3–6 cm s$^{-1}$ alternating (over 12-hr cycles) one of [9]. From the contour plot (Figure 7A), which only shows PAR and food results due to the complexity of visualizing all three culture parameters, it appears that maximum SGR would occur between 200 and 300 μmol m$^{-2}$ s$^{-1}$ and an *Artemia* supply of 30–40 *Artemia* mL$^{-1}$. When factoring in flow, as well, in a prediction profiler based on a least squares model (Figure 7B), the optimal SGR would be predicted to be obtained at a PAR of 290 μmol m$^{-2}$ s$^{-1}$, a food supply of 33 *Artemia* mL$^{-1}$, and a flow regime in which a flow of 3 cm s$^{-1}$ was employed for half of each day and a flow of 6 cm s$^{-1}$ for the other half. The conditions needed to optimize TLE (plots not shown) are similar: PAR, food, and flow of 337 μmol m$^{-2}$ s$^{-1}$, 37 *Artemia* mL$^{-1}$, and alternating flow would lead to a theoretical max. annual increase in TLE of ~10 cm year$^{-1}$. What this means is that the flow regime of our prior work [9], light levels intermediate between the highest of our prior work (250 μmol m$^{-2}$ s$^{-1}$) and lowest of the current one (370 μmol m$^{-2}$ s$^{-1}$), and food levels intermediate between the lowest of the current work (33 *Artemia* mL$^{-1}$) and the highest of the prior one (43 *Artemia* mL$^{-1}$) would be predicted to result in the fastest growing corals; whether or not the theoretical maximum SGR and TLE of 0.8% day$^{-1}$ and 10 cm year$^{-1}$, respectively, could actually be achieved would, of course, need to be validated in future experiments.

The growth rates documented in our RAS with the external feeding tank (TLE = ~6 cm year$^{-1}$) are approximately three-fold higher than those of these corals at the collection site (~2 cm year$^{-1}$ [48]), as well as off Lizard Island, Australia (2.2 cm year$^{-1}$ [49]). They are also higher than those measured *in situ* in the Tropical Eastern Pacific: 2.2–4.6 cm year$^{-1}$ [50]. Richmond [51] found that rates vary considerably over the range of *P. damicornis* (& likely its closely related sister species *P. acuta*), with values ranging from 1.5 to 6 cm year$^{-1}$ at various sites throughout the greater Pacific Rim. In another comparison of field (*in situ*) vs. laboratory (*ex situ*) coral culture data, it is worth highlighting that the opposite trend was documented: *Acropora palmata* nubbins grew significantly more slowly in aquaria than on the reef [52]. Other Taiwanese pocilloporids (*Seriatopora hystrix*) grown in captivity instead exhibited a statistically similar phenotype in NMMBA's husbandry facility as on the local coral reefs of Southern Taiwan [53]. As coral culture and environmental conditions differed across these disparate studies, it is currently premature to propose that the success of *ex situ* coral culture is species-specific, though the degree to which a genotype can be cultured at high growth rates over long-term timescales *ex situ* should certainly be addressed in a larger number of species than the handful that have been the focus of prior studies.

*4.3. Wound Healing*

Counsell et al. [54] reported that, of 108 experimentally wounded *Pocillopora meandrina* samples, 103 (95%) healed within three months (21–86 days), to the extent that it was difficult to distinguish which branches had been broken. Furthermore, larger colonies healed on average 14 days faster than smaller ones. While smaller pocilloporids tend to grow faster than larger colonies [55], the positive relationship between colony size and wound healing speed has been documented in other coral species [56]. Herein all

experimentally wounded corals healed to completion after 28 days, and there was no effect of either light level of feeding regime. In contrast, temperature is likely to affect the wound-healing capacity of this species, as Traylor-Knowles [57] did not observe complete recovery in pocilloporids inhabiting extremely warm waters.

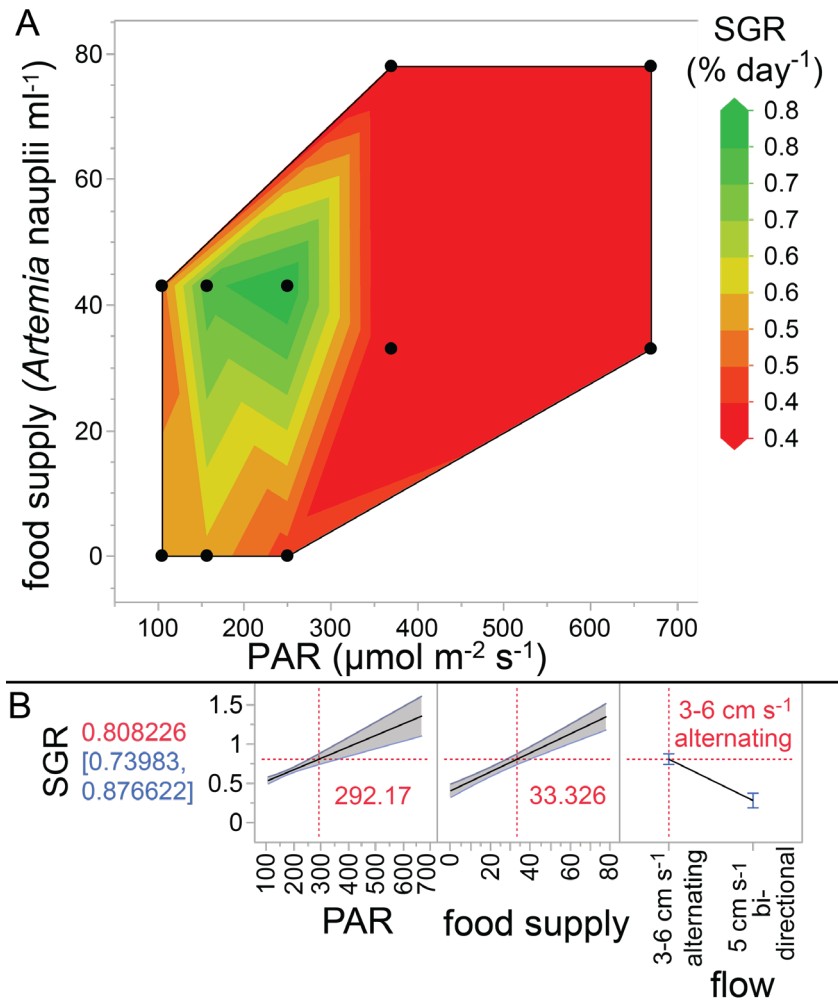

**Figure 7.** Optimizing the *ex situ* culture of *Pocillopora acuta*. Specific growth rate (SGR) data from a prior work with this coral [9] were compared with those obtained herein to attempt to determine the optimal conditions for *P. acuta* growth *ex situ*. Results pooled across the two flow regimes are shown in (**A**), while the specific conditions resulting in the theoretical maximum SGR of 0.8% day$^{-1}$ are shown in (**B**). The values spanning the theoretical maximum SGR (*y*-axis), as well as the error bands and bars in the individual plots, represent 95% confidence, and the optimal photosynthetically active radiation (PAR) levels (as μmol photons m$^{-2}$ s$^{-1}$), food regimes (as *Artemia* mL$^{-1}$), and flow regimes are depicted in the plots within panel (**B**). Note that the approximate maximum value deduced from panel (**A**) differs somewhat from those in (**B**) since the former depicts the mean model across all flow regimes.

### 4.4. RAS-Related Coral Culture Issues and Future Directions

Although all four of the treatment conditions employed resulted in fast-growing corals, it is important to mention one issue with using an external feeding take (aside from the significant increase in human labor required to manually move corals back & forth): the gradual build-up of nutrients in the tank over the duration of the experiment, which we hypothesized caused many corals to begin to undergo tissue sloughing by the 98th day of the experiment (Figure 5B). Despite using a common feeding tank, this presumably nutrient-associated stress event did not affect all corals equally, and those of tank 2 were significantly more compromised relative to those of tank 1, in which no corals were affected.

It is worth mentioning that tissue necrosis in those nubbins of tank 2 and, to a lesser extent, tank 3, began in the hours following feeding, and nitrate levels in the feeding tank were well above those found to induce stress in this coral species in prior experiments [54]. Although the feeding tank was routinely cleaned and flushed with fresh seawater over the course of the experiment, this potentially food-related eutrophication represents one drawback of the RAS. In the future, we plan to use robotic arms to move the nubbins to and from the feeding tank; such automation will minimize the potential for human error and ideally ensure that corals can be cultured successfully *ex situ* for many months, and perhaps even years. Additionally, the addition of a bioreactor (e.g., BioReact 150, Reef Octopus, Manila, Philippines) and a macroalgae reactor (e.g., MBR127, Skimz, Singapore) containing *Chaetomorpha linum* could be added to help eliminate organic wastes more efficiently [58], and we recommend their use in future long-term coral husbandry efforts [59].

**Supplementary Materials:** The following supporting information can be downloaded at https://www.mdpi.com/article/10.3390/oceans5020009/s1.

**Author Contributions:** Conceptualization, T.-Y.F.; methodology, T.-Y.F. and Y.-L.H.; software, T.-Y.F., Y.-L.H. and A.M.; validation, T.-Y.F., Y.-L.H. and A.M.; formal analysis, T.-Y.F., Y.-L.H. and A.M.; investigation, Y.-L.H.; resources, T.-Y.F.; data curation, A.M.; writing—original draft preparation, T.-Y.F.; writing—review and editing, T.-Y.F., Y.-L.H. and A.M.; visualization, T.-Y.F., Y.-L.H. and A.M.; supervision, T.-Y.F.; project administration, T.-Y.F.; funding acquisition, T.-Y.F. All authors have read and agreed to the published version of the manuscript.

**Funding:** This project was funded by the Ministry of Science and Technology (MOST) of Taiwan (MOST 107-2611-M-291-004 to TYF).

**Institutional Review Board Statement:** Animals involved in the experiments are not listed in CITES and were cultured in the laboratory for experimental purposes only. A permit issued to T.-Y.F. for coral collection is referenced in a prior work (upon which this one is based [9]).

**Data Availability Statement:** All data analyzed have been submitted as part of Supplementary File S2. For assistance with reproducing statistical models in JMP$^{®}$ Pro, please contact anderson@coralreefdiagnostics.com or anderson.mayfield@kauat.edu.sa.

**Acknowledgments:** We thank Zong-Min Ye for his comments on the culture systems. We also appreciate valuable comments from Crystal McRae that significantly improved the manuscript. A.B.M. thanks the Fulbright and MacArthur Foundations for funding his stay in Taiwan.

**Conflicts of Interest:** The authors declare no conflicts of interest that may be perceived as inappropriately influencing the representation or interpretation of reported research results. The funders had no role in the design of the study; in the collection, analyses, or interpretation of data; in the writing of the manuscript; or in the decision to publish the results.

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
