# Peer review of "Effects of Food Concentration and Light Intensity on the Growth of a Model Coral"

_2673-1924, doi:10.3390/oceans5020009_

Round 1

Reviewer 1 Report

Comments and Suggestions for Authors

Line 15,16: Four separate conditions that corals were kept in are stated, then a subset of the corals are wounded, making 8 total conditions. But only one result is given (a growth rate/linear extension rate L19)?

Is this because neither light, nor the feeding regime, nor wounding significantly affected the growth rate of the coral? 

L118: SW was changed biweekly? That doesn’t seem like enough. Especially for feeding tank. Increasing nutrients argues that the feeding tank should be changed/cleaned (scrubbed out) after each feeding. 

Pocillopora are hardy corals. There is something going on in the aquaria that are holding them.

The conclusions (that feeding in separate tank-already published (9) and bio-filtration is important) are not enough for publication.

There is another study that was published earlier this year (Gantt et al. 2023, Coral Reefs 42:299-310) that found that corals maintained in aquaria were quite different from the same species on the reef (e.g. lower biomass, lower symbiont density, different symbiont type, higher pigmentation, etc). The message is that if you take corals off the reef, you often (always?) get inferior corals. 

It would be interesting to look at any of parameters in Pocillopora held in aquaria.

Author Response

Reviewer#1 minor comment#1: Line 15,16: Four separate conditions that corals were kept in are stated, then a subset of the corals are wounded, making 8 total conditions. But only one result is given (a growth rate/linear extension rate L19)?

Author response to reviewer #1’s comment#1: This is a good point, and I think much of the confusion can be cleared up by giving the two experiments (which were undertaken under the same conditions but at different times) different names: growth experiment (n=60; non-wounded corals cultured over 120 days) and wounding experiment (n=20 wounded fragments). We have now tried to emphasize when we are talking about the results of each experiment. In the statement in question, we are ONLY talking about growth of the unwounded corals (pooled across the four treatments), and we have modified the sentence to emphasize this.

Reviewer#1 minor comment#2: Is this because neither light, nor the feeding regime, nor wounding significantly affected the growth rate of the coral? 

Author response to reviewer#1’s minor comment#2: Yes, exactly. To be honest, we hypothesized that there would be STRONG effects of light and food, but since this was undocumented, we opted to provide the pooled result across all treatments in the abstract.

Reviewer#1 major comment#1: L118: SW was changed biweekly? That doesn’t seem like enough. Especially for feeding tank. Increasing nutrients argues that the feeding tank should be changed/cleaned (scrubbed out) after each feeding. Pocillopora are hardy corals. There is something going on in the aquaria that are holding them.

Author response to reviewer#1’s major comment#1: This is a good point and one we struggled with. The nitrate levels were effectively zero in the culture tanks. It was only in the feeding tank that nitrate levels rose. What I suspect is that, because it is a RAS, nitrate may have built up in the hard-to-clean areas. However, this would not explain why only corals of one tank (#2) were affected by this; all corals would be immersed in the high-nitrate water during feeding at later experimental data. Therefore, what may have happened is that the high nitrate levels elicited dysbiosis in a random nubbin of the second tank, which than suffered from an infection, which then spread to other nubbins within that tank. It is also possible that there is no relationship with the high-nitrate levels and the tissue necrosis; maybe corals of tank 2 simply became infected. For this reason we generally only discuss the findings when corals were still healthy. I would not put too much stock into these necrosis data; they are more fodder to let people know the kinds of issues they may face if they use RAS (i.e., a caveat to the system). If it’s too distracting to have these observations in the article, then perhaps it would be best to remove them.

Reviewer#1 major comment#2: The conclusions (that feeding in separate tank-already published (9) and bio-filtration is important) are not enough for publication.

Author response to reviewer#1’s major comment#2: Thank you for taking the time to critically review this week, and we apologize for not having better showcased its value, particularly in our failure to distinguish it from its companion article. As mentioned in our response to the editor, we had aimed to do two things: improve the growth rates over our prior work and determine whether certain light levels and feed regimes improve wound recovering. Regarding the latter, all corals recovered quickly, a testament to your noting of the hardiness of this species. Regarding growth, the rates we documented are no better than those of our prior work. Although this is disappointing since it means our hypothesis was unsupported, we do think that these findings can be used to optimize coral husbandry ex situ by effectively showing that certain light and food levels are saturating (“overkill”). This observation could help others in the choosing of optimal light and food regimes for their pocilloporid corals.

Reviewer#1 major comment#3: There is another study that was published earlier this year (Gantt et al. 2023, Coral Reefs 42:299-310) that found that corals maintained in aquaria were quite different from the same species on the reef (e.g. lower biomass, lower symbiont density, different symbiont type, higher pigmentation, etc). The message is that if you take corals off the reef, you often (always?) get inferior corals. It would be interesting to look at any of parameters in Pocillopora held in aquaria.

Author response to reviewer#1 major comment#3: This is a good point and actually frames much of our discussion because we found the OPPOSITE: ex situ growth rates are actually HIGHER than in situ ones. We have some other publications on this with other pocilloporid corals in fact, i.e., in situ vs. ex situ physiological comparisons (Mayfield et al., 2013 J Mar Biol is one such example.). Had we NOT found light or food effects on growth or wound healing AND growth rates were lower than this coral in nearby Nanwan Bay, Taiwan, it is actually likely that we would NOT have submitted the article for publication. In other words, the very high growth rates, in our mind, are the main “selling point” of this article. As such, we have expanded on this section in the discussion to talk more generally about successful coral culture and ex situ vs. in situ differences among the few species tested to date. Hopefully, this helps add a more novel element to the article.

Reviewer 2 Report

Comments and Suggestions for Authors

This study from Fan et al. provides useful information on the physiology of Pocillopora acuta for those wanting to culture the species ex situ, highlighting the importance of light level, and feeding. However, the methods are hard to follow, and the discussion is quite disjointed.

Major comments:

Title: Title is slightly misleading. As only calcification and photosynthetic efficiency were tested, I suggest changing “physiological performance” to “growth and photosynthetic efficiency” or “calcification and photosynthetic efficiency”.

Introduction:

Authors call the coral species a “model coral”, but it is not explained why this species can be called such (especially as many coral species respond differently to environmental conditions) – why is P. acuta a “model coral”? This should be explained early on. If this is not explained, then this should also be changed in the title from “a model coral” to “Pocillopora acuta”.

Methods:

I found the methods extremely hard to follow (see comments below), so believe it would benefit from extensive revision and the addition of an experimental schematic indicating the treatments, tanks, and replicates within each.

Lines 85-87: Do you mean cutting off a branch? When looking at Figure 2, it appears that an entire branch has been removed, not just a “polyp” as authors state here. Please clarify.

Lines 89-90: “…(n=5 corals treatment‐1 ; conducted at a separate time from the longer experiment described below).” – this introduces some confusion over timeline. When was this done and when was the longer experiment done?

For the coral wounding, I understand that 5 wounded nubbins were places into each of four treatments (as there were 20 wounded corals total), with the four treatments being fully factorial with two light levels and two feeding levels. However, I am unclear whether there were replicate tanks for each treatment. In line 100, authors talk of the “three culture tanks”, so clarity is required. An experimental tank schematic would be useful – this would make clear what was in each of the tanks and could also show the upper culture tank and lower life support for each.

How were the sixty non-wounded nubbins distributed across the treatments? Was there even representation of the six starting colonies across the treatments?

Lines 124-131: This is very confusing – 60 fragments were assigned to one of the three tanks. But there were only two light levels, so what is happening in the third tank? Or are the three “culture tanks” replicates of each light level? From my understanding, three “culture tanks” were kept at high light and three at low light (resulting in 6 tanks total, 3 replicates of each light level) with 10 nubbins in each tank, half of which (n=5) were assigned to each feeding regime? However, this is very convoluted in the text. I suggest rewording to something similar to how I worded it above and suggest adding in an experimental schematic for clarity.

Line 148-149: In the statistical analyses section authors state “… (n=five nubbins for each of two PAR levels for each of two feeding regimes in each of three tanks).” – this adds to the confusion. Suggest removing.

Lines 177-178: “To accommodate the split-plot design, whereby multiple PAR levels were employed within the same tank…” – this is the first time at which a split-plot design is introduced within the manuscript and confuses me on the experimental design even further. I understand that the split-plot design explained in the Huang et al. (2020) paper, but it should be mentioned briefly earlier in the methods section here so that the reader can fully understand the experimental design here without having to fully read the other paper. Clarity needs to be added as to how the “split-plot” design was deployed – does this mean that each of the three “culture tanks” employed two light levels?

At what time-points were calcification/growth and photosynthesis measurements made?

Results:

Figure 1: Why is the date displayed on A-C, but experiment day used for D? I suggest using “experimental day” for all.

Figure 1 caption (lines 222-223): “…the means of the mean concentrations…” – is this what you meant to say? Please clarify.

Lines 190-193: Can you provide the average (plus/minus) for the Artemia concentrations here, as you did for the light levels previously?

Lines 196-197: I don’t understand how the nitrate concentration of the feeding tank increased over the course of the study; was this because recirculation was used and so it just built up over time rather than being removed?

Lines 241-248: Can you be sure that the tissue sloughing was due to nitrate concentrations? I think you need to provide a Figure of the nutrient concentration measurements over time in the supplement. I suggest changing Figure S1 to line graphs of the measurements (y-axis) over time (x-axis) with a different coloured line for each tank (1-3).

Discussion:

Discussion is quite disjointed as it stands. I suggest breaking the discussion into sections similar to the methods section, with “coral wounding” as a subheading under which that is discussed, and “coral calcification/growth and photosynthetic efficiency” under which that is discussed, each in the context of light and feeding.

Currently, while FvFm is presented in the results, this is not referenced at all in the discussion – some discussion around this should be added in.

Line 256: “…marginally statistically significant…” – I would suggest calling this a trend rather than marginal significance.

Lines 296-298: This has also been suggested in many Grottoli papers with regards to bleaching – it would be good to cite some here.

Lines 282-302: While it is useful to compare this study to those which have used other species, it would also be useful to compare to Dobson et al. (2021) which found an effect of low light on Pocillopora damicornis in a short timeframe (unable to maintain calcification under low light when fed). I believe that no treatment effect of light was observed here because as you stated earlier, the lower light level used in your study is sufficient for Pocilloporids (lines 264-268), but had you used lower light levels, then you may have seen an effect of light.

Figure 2: This figure needs to be much neater. Each of the picture should be the same size, with smaller text for both the pictures and the graph.

Figure 2 caption: This is the first time that “control” tanks are introduced. These should be introduced and outlined in the methods section. What were the conditions of the control?

Table 2 can go in the supplementary materials.

Figures 3 and 4 could be merged – or it would make more sense to me to have BW and TLE in the same figure, and FvFm in a separate figure. Or, have them as 3 separate figures.

Figure 4: The lower-case letters on panels E and F make it quite cluttered and overwhelming, and I am not sure they really add much to the story. I suggest removing.

Lines 336-344: I am not sure the necrosis can be attributed to the nitrate levels when it affected nubbins from tank 2 more than tanks 1 and 3. Could it be related to the colonies they originally came from? Was that tracked? Was there anything different in tank 2 compared to tanks 1 and 3 that could have caused the necrosis instead?

I don’t understand how the levels of nitrate got so high in the feeding tank when it is stated that the communal feeding tank was routinely cleaned and flushed with fresh seawater over the experiment – is this known? Perhaps it should have been cleaned and flushed between feedings?

Minor comments:

Authors alternate between calling the corals “fragments” and “nubbins” – suggest picking one and using throughout for consistency.

Line 22: Should read “…, and that bio-filtration…”.

Line 45: Suggest swapping “work” for “study”.

Line 47-48: Dobson et al. (2021) also found Pocillopora damicornis required feeding to maintain metabolic demand (at both ambient and elevated temperature). Suggest mentioning this here, too.

Line 58: Suggest calling this “Artemia ml-1” rather than “individuals (ind) ml-1”; as is, this could be confused for coral individuals. Change throughout.

Line 78: Was this artificial or natural seawater? Should be clarified.

Line 78: Should be a space between the number and °C. Should also be a space between numbers/words and ±. Correct throughout.

Line 83: Change from “…weeks at the conditions…” to “…weeks under the conditions…”.

Lines 113-114: How was temperature maintained? Using the osmoregulatory?

Line 118: “…partially synthetic seawater…” – while this is outlined in Huang et al. (2020), I think further comment on this should be added in the main text here by adding in a brief comment on how much was synthetic and how much wasn’t (i.e., 50% synthetic/artificial seawater, 50% natural seawater?).

Line 125: Should be “…were randomly assigned”.

Line 148: Treatment codes need to be spelled out (i.e., high light, high feeding = HLHF; high light, low feeding = HLLF… etc.).

Lines 168-169: Suggest changing to “…presented color values of 5…”.

Line 179: “Tank (treatment)…” – suggest just changing to “treatment” to avoid confusion. However, why would you assess treatment as a random effect?

Line 187: “…at the tank depths…” – what do you mean by this? Clarify needed.

Line 255: “~50% higher” – suggest changing to “1.4x higher”.

Line 266: Suggest changing “recommendable” to “recommended”.

Lines 264-268: This is reasonable, and agrees with findings from Dobson et al. (2021) which found that at low light (150 µmol m-2 s-1), Pocilloporids were not able to maintain calcification even when fed.

Lines 271-273: “When looking at one dimension only (length), nubbins would, on

average, be projected to grow 2.1, 2.1, 3.4, and 2.8 cm per year, respectively (TLE of 6.5, 6.5, 8.7, & 8.2 cm year‐1, respectively).” – it is unclear what the authors mean by this. Please add clarity here.

Lines 279-280: Again, you could also make a comparison to Dobson et al. (2021) here, as they measured calcification under fed/unfed, and high/low light conditions.

Line 336: Should be “required”.

Author Response

Reviewer#2

Reviewer#2 summary: This study from Fan et al. provides useful information on the physiology of Pocillopora acuta for those wanting to culture the species ex situ, highlighting the importance of light level, and feeding. However, the methods are hard to follow, and the discussion is quite disjointed.

Author response to reviewer#2’s summary: Thank you for taking the time to constructively and critically review our article, and we sincerely appreciate your feedback. Upon having re-read it after many months, we can certainly see many of the confusing points in both the methods and elsewhere and have attempted to address these. Of note, we employed a split-plot design, though this was not made very clear. I think that, upon better explaining the three-tank design, in which each tank featured different light-level regimes, the approach will be clearer.

A bigger concern, which was raised by the other reviewers, as well as the editor, is our failure to show any sort of distinction or improvement upon our prior work. Basically, we had attempted to “out-perform” our previously documented growth rates AND demonstrate that wound healing would be affected by light and food. Neither hypothesis was supported, so we have taken a new approach of trying to integrate these new data with those of our prior study to attempt to determine the optimal culture conditions for this coral species. This has also hopefully aided in reducing some of the confusion, of which some derived from not clearly explaining that we had two overarching goals, each addressed by two experiments….which were conducted at different time though using the same culture conditions. Hopefully, upon having addressed these key points, as well as those you have outlined below, this article will be suitable for publication in Oceans’ special edition on coral reefs.

Major comments:

Reviewer#2 comment #1: Title: Title is slightly misleading. As only calcification and photosynthetic efficiency were tested, I suggest changing “physiological performance” to “growth and photosynthetic efficiency” or “calcification and photosynthetic efficiency”.

Author response to reviewer#2’s comment#1: This is a good suggestion, as we did not comprehensively profile the physiological response of this coral. We have opted to simply change it to “growth” since we barely touch on the (uninteresting) PAM (Fv/Fm) data.

Introduction:

Reviewer#2 comment#2: Authors call the coral species a “model coral”, but it is not explained why this species can be called such (especially as many coral species respond differently to environmental conditions) – why is P. acuta a “model coral”? This should be explained early on. If this is not explained, then this should also be changed in the title from “a model coral” to “Pocillopora acuta”.

Author response to reviewer#2’s comment#2: This is because P. acuta was previously synonymized with the world’s best studied coral, Pocillopora damicornis, long touted as the “model coral.” However, since it is not THE only model coral, we refer to it as “a model coral” rather than “THE model coral.” A testament as its model in ecophysiological research is the large number of cited references on this species, particularly in SE Asia. We cited a few references after referring to it as a “model” to emphasize this in the Introduction.

Methods:

Reviewer#2 comment#3: I found the methods extremely hard to follow (see comments below), so believe it would benefit from extensive revision and the addition of an experimental schematic indicating the treatments, tanks, and replicates within each.

Author response to reviewer#2 comment #3: We apologize for the confusing nature of the methods and have attempted to address this by giving the two separate experiments distinct names-growth experiment and wounding experiment-such that the reader can better keep track of which is which. We had also poorly explained the fact that each tank was separated into two “regions” with different light levels and have tried to more clearly emphasize this. It has large implications for how the statistics were done, as it necessitated what is known as a split-plot design.

Reviewer#2 comment#4: Lines 85-87: Do you mean cutting off a branch? When looking at Figure 2, it appears that an entire branch has been removed, not just a “polyp” as authors state here. Please clarify.

Author response to reviewer#2’s comment#4: This is a good catch and, from looking at the pictures, more like 20-30 polyps were raked/scraped off, not a single polyp! We have changed the text to better reflect this.

Reviewer#2’s comment#5: Lines 89-90: “…(n=5 corals treatment‐1 ; conducted at a separate time from the longer experiment described below).” – this introduces some confusion over timeline. When was this done and when was the longer experiment done?

Author response to reviewer#2’s comment#5: This was a mistake, as the experiments did actually occur at the same time: 60 nubbins were simply incubated at each of the four conditions while the other 20 were experimentally wounded first. Aside from the wounding, the major difference between the former “growth experiment” (n=60) and the latter “wound-healing experiment” (n=20) is that the nubbins of the latter were basically only monitored via photography, and neither weighed nor analyzed via PAM. Since there were three tanks, each with two light levels, it would be impossible to have the 20 wounded corals equally divided across the three tanks; instead, there were 4, 6, and 10 in tanks 1, 2, and 3, respectively, with unequal replication across the four

treatments (i.e., it was NOT n=5/treatment for the wounded samples, as had been incorrectly stated and thereby likely cause for your confusion). This error has been corrected. For the unwounded nubbins, there were 20 in each of three tanks; half (n=10/tank) were at the low light level and half were at the high one. Of the 10 corals at each of the light levels in each tank, half (n=5) were given the high-food diet and the other half were given the low-food one. Hopefully, this is now clearer in the revised methods.

Reviewer#2’s comment#5: For the coral wounding, I understand that 5 wounded nubbins were places into each of four treatments (as there were 20 wounded corals total), with the four treatments being fully factorial with two light levels and two feeding levels. However, I am unclear whether there were replicate tanks for each treatment. In line 100, authors talk of the “three culture tanks”, so clarity is required. An experimental tank schematic would be useful – this would make clear what was in each of the tanks and could also show the upper culture tank and lower life support for each.

Author response to reviewer#2’s comment#5: As discussed above, this was actually an error on our part; it would be impossible to equally divide 20 wounded nubbins across three tanks, and so we ended up having unequal sample sizes for this mini-experiment. This would only have been evident had one gone into the supplemental data file. Perhaps this unbalanced design, in fact, thwarted our ability to detect treatment effects on wound-healing, though the data in Figure 5 still seem pretty similar in terms of temporal healing trends.

Reviewer#2’s comment#6: How were the sixty non-wounded nubbins distributed across the treatments? Was there even representation of the six starting colonies across the treatments?

Author response to reviewer#2’s comment#6: Hopefully, this is now clear from our prior responses, but just in case, let me explain it another way. Remember that feeding is done in ANOTHER tank. In other words, there is not a fed half of each tank and then an unfed half; instead, the nubbins are removed individually and place in discrete feeding nets within the feeding tank at their designated feeding times. There are 60 corals divided across three tanks=20 corals/tank. Each tank then has a low and a high-light region= 10 corals/light region/tank. Of the 10 corals in each light region, half are given the high-food regime (n=5), and the other half are given the low-food one. I have now explained this in a simpler way in the main text: “As such, each tank featured five nubbins at each of the four treatments: LLLF, LLHF, HLLF, and HLHF (n=5 nubbins/treatment/tank x 4 treatments x 3 culture tanks).

Reviewer#2’s comment#7: Lines 124-131: This is very confusing – 60 fragments were assigned to one of the three tanks. But there were only two light levels, so what is happening in the third tank? Or are the three “culture tanks” replicates of each light level? From my understanding, three “culture tanks” were kept at high light and three at low light (resulting in 6 tanks total, 3 replicates of each light level) with 10 nubbins in each tank, half of which (n=5) were assigned to each feeding regime? However, this is very convoluted in the text. I suggest rewording to something similar to how I worded it above and suggest adding in an experimental schematic for clarity.

Author’s response to reviewer#2’s comment#7: It was not a 3-high vs. 3-low-light tank design (which is, admittedly, more common); this is because we had so many nubbins and we wanted to space them out. Instead, each tank featured each of the two light levels. Corals were immersed in the same seawater, but the tank lights were optimized to where different regions of the tank experienced different light regimes. This necessitated a split-plot design. Each tank had 10 corals in each of two light regions. Half of each of these 10 were given the high-food diet and half were given the low-food one. So another way to picture it would be: 5 corals/food regime/light region/tank x 2 food regimes x 2 light regions/tank x 3 tanks = 60 total nubbins.

Reviewer#2 comment#8: Line 148-149: In the statistical analyses section authors state “… (n=five nubbins for each of two PAR levels for each of two feeding regimes in each of three tanks).” – this adds to the confusion. Suggest removing.

Author response to reviewer#2’s comment#8: Although this is technically correct, we now state this earlier to attempt to clear up the aforementioned confusion.

Reviewer#2 comment#9: Lines 177-178: “To accommodate the split-plot design, whereby multiple PAR levels were employed within the same tank…” – this is the first time at which a split-plot design is introduced within the manuscript and confuses me on the experimental design even further. I understand that the split-plot design explained in the Huang et al. (2020) paper, but it should be mentioned briefly earlier in the methods section here so that the reader can fully understand the experimental design here without having to fully read the other paper. Clarity needs to be added as to how the “split-plot” design was deployed – does this mean that each of the three “culture tanks” employed two light levels?

Author response to reviewer#2’s comment#9: This is a good suggestion to mention that it is a split-plot earlier in the methods. We actually give an overview of the design in the revised first paragraph of the Methods and mention the split-plot nature now. Hopefully, this will clarify things for the reader. Basically, we found that corals like these large tanks for a potential variety of reasons, so rather than have tons of small tanks, each with a discrete factorial combination of conditions, we use larger tanks, then sub-divide them into “regions.”

Reviewer#2 comment#10: At what time-points were calcification/growth and photosynthesis measurements made?

Author response to reviewer#2’s comment#10: This is a good point, as we had previously relied on the reader to figure this out simply by looking at the figures. I have now directly mentioned this in the Materials and Methods.

Results:

Reviewer#2 comment#11: Figure 1: Why is the date displayed on A-C, but experiment day used for D? I suggest using “experimental day” for all.

Author response to reviewer#2 comment#11: Although we could see why this would be confusing, this is the only part of the article where we actually highlight the actual dates, and these

might be of interest to future readers since there are seasonal differences in coral physiology that must be considered in these multi-month experiments.

Reviewer#2’s comment#12: Figure 1 caption (lines 222-223): “…the means of the mean concentrations…” – is this what you meant to say? Please clarify.

Author response to reviewer#2’s comment#12: We struggled with this sentence because, despite sounding like a mistake, it was actually the mean of means (since so much data were collected, this simplified things).

Reviewer#2’s comment#13: Lines 190-193: Can you provide the average (plus/minus) for the Artemia concentrations here, as you did for the light levels previously?

Author response to reviewer#2’s comment#13: We had originally opted not to do this because these values inherently differ from the “starting values” because, over the course of the 6-hr period, the corals consume the Artemia, hence the value is always dropping. These data are shown in Figure S2. However, we include the mean concentration over the 6-hr period, as that could basically be thought of as how much food a coral would have access to halfway through the feeding event.

Reviewer#2’s comment#14: Lines 196-197: I don’t understand how the nitrate concentration of the feeding tank increased over the course of the study; was this because recirculation was used and so it just built up over time rather than being removed?

Author response to reviewer#2’s comment#14: Honestly, this is the same question I had; it’s not as if we allowed these tanks to fester for weeks. They were cleaned regularly and seawater was exchanged regularly. Furthermore, if the source seawater had issues, it would have affected the culture tanks, yet they were virtually oligotrophic throughout! The only thing I can think of is that food wastes or even nitrogenous wastes from the coral, adhered to the plumbing to where it could not be simply wiped away, akin to plaque buildup on teeth. Suffice to say, this is one major drawback of the RAS, hence why we felt compared to share this finding (which most would probably NOT want to have peer reviewed since it could emphasize that you have no idea what you’re doing!).

Reviewer#2’s comment#15: Lines 241-248: Can you be sure that the tissue sloughing was due to nitrate concentrations? I think you need to provide a Figure of the nutrient concentration measurements over time in the supplement. I suggest changing Figure S1 to line graphs of the measurements (y-axis) over time (x-axis) with a different coloured line for each tank (1-3).

Author response to reviewer #2’s comment#15: We actually do show the nitrate build-up, which was ONLY in the feeding tank (not the coral culture tanks, thankfully). But to answer your first question: no, it was more because tissue sloughing was seen shortly after immersion of corals in the high-nutrient water around day 90. The fact that this phenomenon ONLY affected corals of one tank actually points to it being something beyond simply nutrients, or else all corals would have suffered equally since they all go into the same feeding tank. It would appear that, despite growing at similar rates and presenting normal appearance and Fv/Fm, corals of tank 2 were, for s

some reason, more susceptible to high nitrogen. Probably what happened is that one nubbin of that tank had a low-grade infection that was exacerbated by the high-nitrogen; this disease then spread like wild-fire to other nubbins in that same tank. If this does not appear overly speculative, we could include some of these hypotheses in the text, though we had opted to keep it vague since we honestly have little to go by.

Discussion:

Reviewer #2 comment #16: Discussion is quite disjointed as it stands. I suggest breaking the discussion into sections similar to the methods section, with “coral wounding” as a subheading under which that is discussed, and “coral calcification/growth and photosynthetic efficiency” under which that is discussed, each in the context of light and feeding.

Author response to reviewer#2’s comment#16: This is a good suggestion and you will find that the Discussion has changed considerably as a response to the other reviewers’ comments. Of note, there is a new section on attempting to make projections about optimal coral growth conditions. We now talk about light and food effects on growth (or lack thereof), optimized coral culture (including comparisons to other studies), wound healing, and experimental issues and caveats (in that order). Hopefully, with the sub-headings, it is less likely that the reader will be lost.

Reviewer#2 comment#17: Currently, while FvFm is presented in the results, this is not referenced at all in the discussion – some discussion around this should be added in.

Author response to reviewer#2’s comment#17: Honestly, we only include PAM fluorometry because we felt compelled by the coral field to do so. The results have no bearings on photosynthesis or carbon fixation, but in any event, we saw very few changes in response to experimental treatments. And even though Fv/Fm was, for some reason, significantly higher on day 28 vs. day 42 (Table 2), the magnitude of this difference is along the lines of 1-2% (i.e., not biologically meaningful).

Reviewer#2 comment#18: Line 256: “…marginally statistically significant…” – I would suggest calling this a trend rather than marginal significance.

Author response to reviewer#2’s comment#18: This is a good suggestion that makes it seem less like we are trying to find something that is not actually there. I have reworded the sentence to accommodate this.

Reviewer#2 comment#19: Lines 296-298: This has also been suggested in many Grottoli papers with regards to bleaching – it would be good to cite some here.

Author response to reviewer#2’s comment#19: This is a good suggestion, and one had actually already been cited elsewhere in the article, so I have now cited a second. I really regret NOT sampling for lipid work with these corals, in fact.

Reviewer#2 comment#20: Lines 282-302: While it is useful to compare this study to those which have used other species, it would also be useful to compare to Dobson et al. (2021) which found an effect of low light on Pocillopora damicornis in a short timeframe (unable to maintain

calcification under low light when fed). I believe that no treatment effect of light was observed here because as you stated earlier, the lower light level used in your study is sufficient for Pocilloporids (lines 264-268), but had you used lower light levels, then you may have seen an effect of light.

Author response to reviewer#2’s comment#20: This is a good suggestion, and we have now mentioned this. That being said, we undertook a new analysis (Figure 6) where we attempted to predict the optimal light level for this coral in Taiwan and it was in the low 300 umol/m2/s. Keep in mind that these corals were originally collected at 3-5 m (i.e., shallow), so this finding is completely unsurprising; they are used to high PAR in situ. That being said, 670, while it didn’t elicit stress, is probably “overkill.”

Reviewer#2 comment#21: Figure 2: This figure needs to be much neater. Each of the picture should be the same size, with smaller text for both the pictures and the graph.

Author response to reviewer#2’s comment#21: We can only do so much with this figure since the images are low-resolution thumbnails, and each cannot be individually sized. What this means is that they either have to be mis-aligned OR mis-sized. I am hoping that it is not that noticeable in print. However, we were able to make the text smaller, as that had simply been overlaid.

Reviewer#2 comment#22: Figure 2 caption: This is the first time that “control” tanks are introduced. These should be introduced and outlined in the methods section. What were the conditions of the control?

Author response to reviewer#2 comment#22: This is a good catch, as that is supposed to be “culture” tanks. There were no culture tanks, per se, in this experiment, and I have corrected this.

Reviewer#2 comment#23: Table 2 can go in the supplementary materials.

Author response to reviewer#2 comment#23: Despite this table featuring primarily negative data (i.e., no effect of treatment), we do still wish to emphasize some of the temporal variation and so would opt to leave it in the main text for now. If the editor and other reviewers feel similarly, however, we can easily move it to the online supplement.

Reviewer#2 comment#24: Figures 3 and 4 could be merged – or it would make more sense to me to have BW and TLE in the same figure, and FvFm in a separate figure. Or, have them as 3 separate figures.

Author response to reviewer#2 comment#24: This is a good section, and we have divided them into three figures. This also addresses your next comment, that the Fv/Fm plots are too crowded (new Figure 5). However, since we did not discuss these post-hoc differences at all in the text, I think it is fine to remove them per your suggestion.

Reviewer#2 comment #25: Figure 4: The lower-case letters on panels E and F make it quite cluttered and overwhelming, and I am not sure they really add much to the story. I suggest removing.

Author response to reviewer#2’s comment#25: Please see our response to your prior concern.

Reviewer#2 comment#26: Lines 336-344: I am not sure the necrosis can be attributed to the nitrate levels when it affected nubbins from tank 2 more than tanks 1 and 3. Could it be related to the colonies they originally came from? Was that tracked? Was there anything different in tank 2 compared to tanks 1 and 3 that could have caused the necrosis instead?

I don’t understand how the levels of nitrate got so high in the feeding tank when it is stated that the communal feeding tank was routinely cleaned and flushed with fresh seawater over the experiment – is this known? Perhaps it should have been cleaned and flushed between feedings?

Author comment to reviewer#2’s comment#26: We actually DID know the source colonies, though no one source colony was more susceptible. What I think may have happened is that, despite cleaning and flushing, nitrogenous wastes built up in the plumbing somewhere that could not easily be cleaned. Perhaps these high-nitrogen levels caused dysbiosis of the microbial flora of a random nubbin in tank 2, which then led to an infection, which then spread to other nubbins in that tank. This could explain why it was mainly corals of one tank that were affected, despite them all being exposed to high-nitrate seawater.  We realize that it is unusual to talk about parts of the experiment that did not work, and if this section ends up being too confusing or distracting, we could simply opt to remove it.

Minor comments:

Reviewer#2 minor comment#1: Authors alternate between calling the corals “fragments” and “nubbins” – suggest picking one and using throughout for consistency.

Author response to reviewer#2’s minor comment#1: This is a good section, and we have opted to use “nubbin” since it was the more common of the two.

Reviewer#2 minor comment#2: Line 22: Should read “…, and that bio-filtration…”.

Author response to reviewer#2’s minor comment#2: It actually SHOULD be how it is written, but I can see why this sentence reads strangely, so I broken it into two sentences: “This highlights the importance of feeding in separate tanks during long-term culture of corals; perhaps bio-filtration could reduce the possibility of organic matter accumulation in future coral culture initiatives.”   

Reviewer#2 minor comment# 3: Line 45: Suggest swapping “work” for “study”.

Author response to reviewer#2’s minor comment#3: The suggested change has been made.

Reviewer#2 minor comment#4: Line 47-48: Dobson et al. (2021) also found Pocillopora damicornis required feeding to maintain metabolic demand (at both ambient and elevated temperature). Suggest mentioning this here, too.

Author response to reviewer#2’s minor comment#4: The suggested change has been made.

Reviewer#2 minor comment#5: Line 58: Suggest calling this “Artemia ml-1” rather than “individuals (ind) ml-1”; as is, this could be confused for coral individuals. Change throughout.

Author response to reviewer#2’s minor comment#5: The suggested change has been made.

Reviewer#2 minor comment#6: Line 78: Was this artificial or natural seawater? Should be clarified.

Author response to reviewer#2’s minor comment#6: It was natural in this part of the experiment, and this has now been emphasized.

Reviewer#2 minor comment#7: Line 78: Should be a space between the number and °C. Should also be a space between numbers/words and ±. Correct throughout.

Author response to reviewer#2’s minor comment#7: This is actually untrue, though many get it wrong in publications (especially recently). There should be NO space in these instances, but there SHOULD be a space between numbers and units besides temperature. Not sure why this is the case actually.

Reviewer#2 minor comment#8: Line 83: Change from “…weeks at the conditions…” to “…weeks under the conditions…”.

Author response to reviewer#2 minor comment#8: The suggested change has been made.

Reviewer#2 minor comment#9: Lines 113-114: How was temperature maintained? Using the osmoregulatory?

Author response to reviewer#2’s minor comment #9: It was actually regulated by the heater and chiller described in the prior sentence (each made adjustments as needed if temperature deviated significantly). This has now been mentioned.

Reviewer#2 minor comment#10: Line 118: “…partially synthetic seawater…” – while this is outlined in Huang et al. (2020), I think further comment on this should be added in the main text here by adding in a brief comment on how much was synthetic and how much wasn’t (i.e., 50% synthetic/artificial seawater, 50% natural seawater?).

Author response to reviewer#2 minor comment#10: This is a good point as we had failed to mention this. It is actually the same as in Huang et al. and so is simply “synthetic seawater” (Red Sea Salts+RO water). I have now corrected this.

Reviewer#2 minor comment#11: Line 125: Should be “…were randomly assigned”.

Author response to reviewer#2’s minor comment#11: The sentence in question no longer exists as such since this section was modified dramatically per other reviewer comments.

Reviewer#2 minor comment#12: Line 148: Treatment codes need to be spelled out (i.e., high light, high feeding = HLHF; high light, low feeding = HLLF… etc.).

Author response to reviewer#2’s minor comment#12: The suggested change has been made.  

Reviewer#2’s minor comment#13: Lines 168-169: Suggest changing to “…presented color values of 5…”.

Author response to reviewer#2’s minor comment#13: The suggested change has been made.

Reviewer#2 minor comment#14: Line 179: “Tank (treatment)…” – suggest just changing to “treatment” to avoid confusion. However, why would you assess treatment as a random effect?

Author response to reviewer#2’s minor comment#14: The parentheses refer to a nested effect: “tank nested within treatment.” Since it’s a split-plot, it’s sort of turned on its head, and it’s important to show that tank effects within a given treatment do not bias results. Definitely are not testing treatment as a random effect!

Reviewer#2 minor comment#15: Line 187: “…at the tank depths…” – what do you mean by this? Clarify needed.

Author response to reviewer#2’s minor comment#15: This is an error on my part, as I had thought the light level varied within tanks on account of the nubbin depths when in fact it had to do with light positioning (explained elsewhere). I have corrected this now.  

Reviewer#2 minor comment#16: Line 255: “~50% higher” – suggest changing to “1.4x higher”.

Author response to reviewer#2’s minor comment#16: The suggested change has been made.

Reviewer#2 minor comment#17: Line 266: Suggest changing “recommendable” to “recommended”.

Author response to reviewer#2’s minor comment#17: The suggested change has been made.

Reviewer#2 minor comment#18: Lines 264-268: This is reasonable, and agrees with findings from Dobson et al. (2021) which found that at low light (150 µmol m-2 s-1), Pocilloporids were not able to maintain calcification even when fed.

Author response to reviewer#2’s minor comment#18: Please see elsewhere on how we have made this inter-study comparison.

Reviewer#2 minor comment#19: Lines 271-273: “When looking at one dimension only (length), nubbins would, on average, be projected to grow 2.1, 2.1, 3.4, and 2.8 cm per year, respectively (TLE of 6.5, 6.5, 8.7, & 8.2 cm year‐1, respectively).” – it is unclear what the authors mean by this. Please add clarity here.

Author response to reviewer#2’s minor comment#19: This sentence had all sorts of issues (including some of the data being slightly off). I now make it clear that the 2.1-2.8 values reflect annual increases in LENGTH while the 6.5-8.2 values are TLE, i.e., all dimensions summed.

Reviewer#2 minor comment#20: Lines 279-280: Again, you could also make a comparison to Dobson et al. (2021) here, as they measured calcification under fed/unfed, and high/low light conditions.

Author response to reviewer#2 minor comment#20: The suggested change has been made.

Reviewer#2 minor comment#21: Line 336: Should be “required”. 

Author response to reviewer#2’s minor comment#21: Good catch! The suggested change has been made.

Reviewer 3 Report

Comments and Suggestions for Authors

In this manuscript, authors carried out an investigation on the effects of food concentration and light intensity on the physiological performance of the model reef coral Pocillopora acuta cultured in recirculating aquaculture systems. This highlights the importance of feeding in separate tanks during long-term culture of reef corals. These findings were important, which made this work worth publishing in this journal.

Comments:

1. In the authors’ previous study as recorded in Ref. [9], the effects of feeding on the physiological performance of the stony coral Pocillopora acuta were investigated. However, it was studied again in this study. What was your purpose?

2. The Ref. [15] showed that different ages can have different acclimation to the light levels. Were the six P. acuta colonies used in this study in the same age?

3. Did you detect the profiles of secondary metabolites of the cultured nubbins? The cultivating corals were a sustainable supply of biological material for natural products, some of which were the important sources of pharmaceuticals.

Other revisions:

1. It is better to put the figure captions after the pictures.

2. Please pay attentions to the author lists of the references at the end of this manuscript. In some references, such as [4] and [6], all the authors were given. However, in other references, such as [13] and [28], only one or three authors were put.

Author Response

Reviewer#3 summary: In this manuscript, authors carried out an investigation on the effects of food concentration and light intensity on the physiological performance of the model reef coral Pocillopora acuta cultured in recirculating aquaculture systems. This highlights the importance of feeding in separate tanks during long-term culture of reef corals. These findings were important, which made this work worth publishing in this journal.

Author response to reviewer #3’s summary: Thank you for taking the time out of your busy schedule to critically and constructively review our work, and we are glad that you see its value, despite your concerns outlined below. Your#1 point is, of course, the one echoed by all three reviewers, as well as the editor, and admittedly, is a sort of “elephant in the room” since both of our hypotheses were wrong. First, we thought we could further optimize the light and food regimes to achieve superior growth rates to those of our prior study. We did not. Secondly, we sought to determine light and food regime effects on wound healing, but none were documented. Rather than simply highlight these “negative” findings, we have instead done two things. First, we have highlighted the high growth rates, which are higher than those in situ. Secondly, we carried out a meta-analysis with prior data to estimate the optimal culture conditions for this species (new Figure 6). Hopefully, the latter will emphasize that, through looking at the combined results of both studies, some insight into the optimal culture conditions for this species can be garnered.

Comments:

Reviewer#3 major comment#1. In the authors’ previous study as recorded in Ref. [9], the effects of feeding on the physiological performance of the stony coral Pocillopora acuta were investigated. However, it was studied again in this study. What was your purpose?

Author response to reviewer#3 major comment#1: Please see our response above. Briefly, we had hoped to achieve BETTER results than those of our prior study, as well as identify the optimal conditions for wound healing. Since we could confirm neither, I have now added a new section to highlight the results of a new figure, Figure 6, which presents a meta-analysis of the results of both studies.

Reviewer#3 major comment#2. The Ref. [15] showed that different ages can have different acclimation to the light levels. Were the six P. acuta colonies used in this study in the same age?

Author response to reviewer#3 comment #2: Although we do not know the exact age of the colonies used, since they were obtained from the ocean, we did make an effort to collect similarly sized colonies, and they were relatively small and therefore presumably young (12-15 cm). Indeed, we mention the very fast growth rates of smaller corals in the Discussion, and using differently aged corals would certainly have biased the growth rate data in particular, so this is a good comment.

Reviewer#3 major comment#3. Did you detect the profiles of secondary metabolites of the cultured nubbins? The cultivating corals were a sustainable supply of biological material for natural products, some of which were the important sources of pharmaceuticals.

Author response to reviewer#3’s major comment#3: We did not sample these corals, though this is a good suggestion, and it would have helped us better understand how they were storing energy and whether they were relying more on autotrophy or heterotrophy under the various conditions. We included a sentence in the Discussion to emphasize the need for a metabolomic approach in future such studies.

Minor revisions

Reviewer#3 minor comment#1. It is better to put the figure captions after the pictures.

Author response to reviewer#3 minor comment#1: This is actually a requirement for MDPI, and they will reformat it as such later. However, we prefer to let the reviewer read the description of the figure BEFORE seeing it, so that there is less confusion.

Reviewer#3 minor comment#2. Please pay attentions to the author lists of the references at the end of this manuscript. In some references, such as [4] and [6], all the authors were given. However, in other references, such as [13] and [28], only one or three authors were put.

Author response to reviewer#3 minor comment#2: This is a good catch, and we apologize for the improperly formatted references. We have now done our best to ensure that they are all in the same format (with all authors listed).

Round 2

Reviewer 1 Report

Comments and Suggestions for Authors

None

Author Response

Thank you for taking the time to look at our article again. We hope it is now suitable for publication in Oceans

Reviewer 2 Report

Comments and Suggestions for Authors

The authors have responded well to the reviewer comments and have added a lot of clarification to the methods; it is now a lot clearer.

I have a question about having the two light levels within the same tank; how was the light separated within the same tank? Could some of the higher light ‘leaked’ over into the low light, so the light levels you thought the corals were receiving was slightly different? Was there obvious space between the two “regions”?

Figure 1: I understand your argument for keeping this the same. Could you instead alter the x-axes so that the month day/experiment day line up on A-C and D?

Lines 209-211: For the Artemia concentration, I would instead suggest reporting the initial/starting concentration of the Artemia which was offered to the corals; I think you cite the starting concentration in lines 144-145?

Author Response

Reviewer#2: summary: The authors have responded well to the reviewer comments and have added a lot of clarification to the methods; it is now a lot clearer.

Author response to reviewer#2’s summary: Thank you for taking the time to re-review our significantly revised article. We are pleased that you have now found it easier to read, and we have done our best to address your detailed concerns below.

Reviewer#2 comment#1: I have a question about having the two light levels within the same tank; how was the light separated within the same tank? Could some of the higher light ‘leaked’ over into the low light, so the light levels you thought the corals were receiving was slightly different? Was there obvious space between the two “regions”?

Author response to reviewer#2’s comment#1: This is a good comment/question and showcases that we had failed to describe this properly. There was a screen between the regions, but it did not dip into the water since that would have affected the flow regime. This means that there was likely some “leaking” or “bleeding” effect, in which fragments near the edge may have sometimes received higher light than others of the same light treatment. For this reason, it was important to rotate nubbins randomly around the tank. And in fact, you can see the “bleeding” effect to some extent in Figure 1A; on some measurement days, there was a huge difference (>2-fold) between the low- and high-light treatments (e.g., day 15) whereas on others it was less (e.g., day 20). This also reflects the fact that the exact position of the Licor in each light region of the tank was randomized, in some instance being closer to the border and in other instances closer to the opposing edge (i.e., greater difference in PAR). Since we believe others may have this same question, I have added a sentence to the methods to explain this: “A plastic divider was placed between the lamps, though since it did not enter the water (to avoid interfering with the flow), it is possible that light from one light region of the tank entered the other; for this reason, it was important to randomly rotate nubbins regularly within each tank light region to ensure that those on the low-high-light border did not experience significantly different PAR regimes from other nubbins within the same treatment.”

Reviewer#2 comment #2: Figure 1: I understand your argument for keeping this the same. Could you instead alter the x-axes so that the month day/experiment day line up on A-C and D?

Author response to reviewer#2 comment#2: Since we had (confusingly) shown 90 days of light (PAR) data and 112 days of feeding data, I have opted to simply remove the final ~15 days of the latter so that the scales are roughly similar (i.e., panels A-C in sync with D). Hopefully, this now looks better.  

Reviewer#2 comment#3: Lines 209-211: For the Artemia concentration, I would instead suggest reporting the initial/starting concentration of the Artemia which was offered to the corals; I think you cite the starting concentration in lines 144-145?

Author response to reviewer#2’s comment#3: This is a good point, as we normally opted to present the initial level. I have now added the initial values in parentheses here because in this section, I did want to emphasize that the actual concentration available to a coral over the duration of the feeding period was LESS THAN the starting value, as the corals ate quite voraciously (see Figure S2 to see how fast Artemia levels dropped.). I believe elsewhere we simply refer to the initial values, though I will double-check for consistency.

Reviewer 3 Report

Comments and Suggestions for Authors

The authors have addressed all my concerns; thus, this manuscript can be accepted in the present form.

Author Response

Thank you for your critical review of our manuscript. Your comments, as well as those of the other two reviewers, have significantly improved it and made the message far clearer. 

Round 3

Reviewer 2 Report

Comments and Suggestions for Authors

I am happy with how the authors have amended the manuscript in response to me comments and feel that this is now ready to be published.

Author Response

Thank you for endorsing our work, and your comments, as well as those of the other reviewers, have certainly improved its clarity (which was admittedly muddled in the originally submitted version).